# Living fast in the Triassic: New data on life history in *Lystrosaurus* (Therapsida: Dicynodontia) from northeastern Pangea

Zoe T. Kulik[1]*, Jacqueline K. Lungmus[2], Kenneth D. Angielczyk[3], Christian A. Sidor[1]

**1** Department of Biology and Burke Museum, University of Washington, Seattle, WA, United States of America, **2** Smithsonian National Museum of Natural History, Washington, D.C., United States of America, **3** Negaunee Integrative Research Center, Field Museum of Natural History, Chicago, IL, United States of America

\* zkulik@uw.edu

**Data Availability Statement:** All relevant data are within the manuscript and its Supporting Information files. Thin section images are available

## Abstract

*Lystrosaurus* was one of the few tetrapods to survive the Permo-Triassic mass extinction, the most profound biotic crisis in Earth's history. The wide paleolatitudinal range and high abundance of *Lystrosaurus* during the Early Triassic provide a unique opportunity to investigate changes in growth dynamics and longevity following the mass extinction, yet most studies have focused only on species that lived in the southern hemisphere. Here, we present the long bone histology from twenty *Lystrosaurus* skeletal elements spanning a range of sizes that were collected in the Jiucaiyuan Formation of northwestern China. In addition, we compare the average body size of northern and southern Pangean Triassic-aged species and conduct cranial geometric morphometric analyses of southern and northern taxa to begin investigating whether specimens from China are likely to be taxonomically distinct from South African specimens. We demonstrate that *Lystrosaurus* from China have larger average body sizes than their southern Pangean relatives and that their cranial morphologies are distinctive. The osteohistological examination revealed sustained, rapid osteogenesis punctuated by growth marks in some, but not all, immature individuals from China. We find that the osteohistology of Chinese *Lystrosaurus* shares a similar growth pattern with South African species that show sustained growth until death. However, bone growth arrests more frequently in the Chinese sample. Nevertheless, none of the long bones sampled here indicate that maximum or asymptotic size was reached, suggesting that the maximum size of *Lystrosaurus* from the Jiucaiyuan Formation remains unknown.

## Introduction

The Permo-Triassic mass extinction was the most devastating biotic crisis in Earth's history and caused communities to collapse in both the terrestrial and marine realms [1–3]. Fluctuating climates accompanied by an overall increase in global temperatures are hypothesized to have forced 70% of terrestrial vertebrate families to extinction [4–8], although plant extinctions were notably damped [9–11]. Recovery after the Permo-Triassic extinction has been a topic of

on Morphobank at project number P4023 (http://morphobank.org/permalink/?P4023).

**Funding:** This research was supported by National Science Foundation grants EAR 1714829 (to KDA) and EAR 1713787 (to CAS), and UW Biology Department Iuvo and Walker awards (to ZTK). The funders had no role in study design, data collection and analysis, decision to publish, or preparation of the manuscript.

**Competing interests:** The authors have declared that no competing interests exist.

intense research, but most studies in the terrestrial realm focus on taxonomic diversity and abundance [12–14], or changes in community structure and dynamics following the extinction [6, 3, 15–18]. Few studies have examined how the extremely changed environment of the earliest Triassic impacted growth rate, life history, and longevity of surviving taxa [19–22]. Importantly, most of the studies noted above rely heavily—if not exclusively—on data from the southern hemisphere, in particular the Karoo Basin of South Africa.

*Lystrosaurus*, a stoutly built dicynodont therapsid that ranged in maximum skull size from approximately 16–39 cm (*L. murrayi* and *L. maccaigi* from South Africa, respectively), is one of the hallmark survivors of the Permo-Triassic extinction due to its remarkable abundance in southern Pangean deposits [18, 23, 24]. The success of *Lystrosaurus* has been variously attributed to its generalist diet of tough plant material (e.g., [25, 26]), broad habitat tolerances [27], a burrowing lifestyle [28], unusual thermal tolerances [29], and a developmentally plastic growth strategy that allowed it to weather extreme ecosystem instability [20, 22, 30, 31]. These hypothesized advantages/exaptations for a post-extinction environment have been studied almost exclusively using specimens from South Africa, but *Lystrosaurus* is known from across Pangea: including Permo-Triassic aged deposits in Antarctica, Mongolia, Russia, China, India, and possibly Mozambique (e.g., [32–39]). The wide paleolatitudinal range of *Lystrosaurus* offers a rare opportunity to investigate differences in lifespan, survivorship, and body size in southern and northern Pangean populations that experienced markedly different environmental conditions in the wake of the Permo-Triassic extinction (e.g., [22, 40]).

## Osteohistological perspectives on life history

Bone histology provides a wealth of information about an extinct animal's growth and life history. Mineralized bone matrix, vascular canal spaces, and cellular spaces persist in fossilized bone, making it possible to determine the relative rate of bone deposition as well as maturity of extinct organisms [41]. Cyclical decreases in skeletal growth rate leave histological markers, termed growth marks, in primary cortical bone. Annuli and lines of arrested growth (LAGs) are two types of cyclical growth marks that are deposited in annual or seasonal cycles [42–44]. Annuli represent periodic decreases in growth rate and are usually identified based on specific, contextual changes in the bone tissue composition, such as a dramatic decrease of vascular and cellular density, or a temporary shift to parallel-fibered or lamellar bone. LAGs, on the other hand, represent periodic cessations in growth and appear as hyper-mineralized lines in the primary cortical bone of most vertebrates when bone deposition stops in unfavorable seasons [42, 44–49]. In addition to extrinsically (i.e., environmentally or seasonally) induced growth marks, growth can also arrest during periods of intense stress, such as birth, and during modulations to metabolic activity such as torpor [50, 51]. The periodicity of growth marks (i.e., the extent to which they are regularly spaced throughout the cortex of large, presumably mature individuals) helps to determine whether they represent extrinsic or intrinsic events.

Botha [22] recently conducted a detailed examination of growth and life history patterns in four species of *Lystrosaurus* across the Permian-Triassic boundary using fossils from South Africa. Using a combination of bone histology and body size data, Botha [22] found that Triassic *Lystrosaurus* individuals grew rapidly and had increased mortality at small sizes, as evidenced by the overwhelming abundance of small skulls known from Triassic rocks in the Karoo Basin. This post-extinction change in life expectancy is also inferred from the scarcity of LAGs and annuli in the rapidly deposited cortical bone of Triassic specimens. In comparison, Permian individuals had larger average skull sizes and had numerous growth marks spaced throughout the cortex, suggesting prolonged periods of growth spanning multiple seasons or years [22]. Similar patterns were previously found across a wider taxonomic range of South

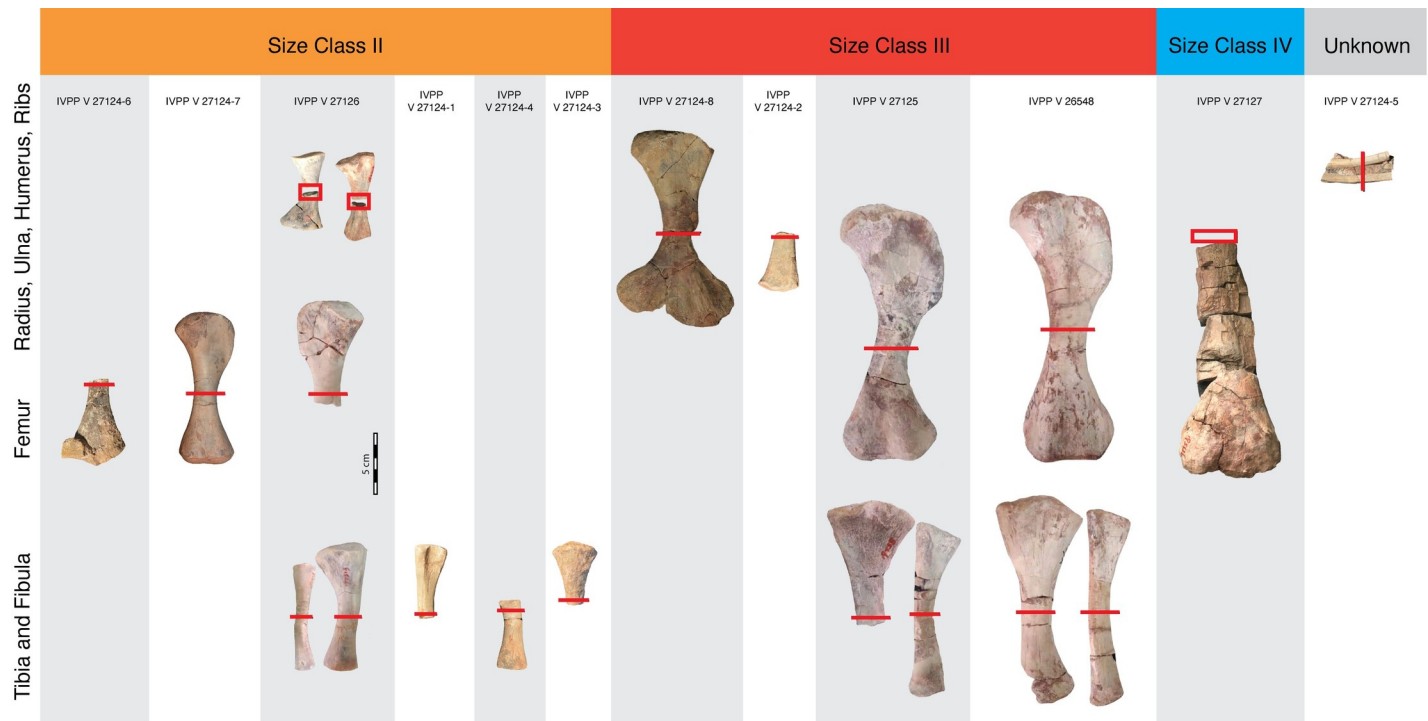

**Fig 1. Size distribution of *Lystrosaurus* sp. postcrania from the Jiucaiyuan Formation included in the histologic analysis.** Twenty skeletal elements, with gray bars indicating associations of individuals, span three size classes inferred from femoral length: Size Class II spans 47–63% maximum femoral size, Size Class III includes specimens that are ~ 70% maximum size, and Size Class IV includes the largest partial femur. Scale bar = 5 cm.

African therapsid species [20] and lydekkerinid stereospondyls [52, 53]. The predominance of small *Lystrosaurus* (and other tetrapods) with rapidly deposited bone tissue suggests an overall shift in life history strategy across the Permian-Triassic boundary to an early onset of reproductive maturity [20, 52, 53].

Histological data for *Lystrosaurus* outside of the South African Karoo Basin are more limited, but critical to understanding the relationship between latitude (and by extension climate) and life history. *Lystrosaurus* from the Early Triassic of India lack LAGs and annuli, and have small skulls, consistent with the pattern seen in South Africa [37, 54]. A preliminary report of the bone histology of *Lystrosaurus* from the Turpan-Hami Basin of China suggested that subadult individuals record multiple LAGs [40]. However, in the absence of comparative body size data, it remains unclear if multiple LAGs in the Chinese specimens indicate prolonged periods of growth (i.e., multiple cycles or seasons of growth) or environmental instability (e.g., disruptions to the metabolic rate as a result of resource limitations). Here, we analyze bone histology from a size range of *Lystrosaurus* postcrania from the Jiucaiyuan Formation of Xinjiang, China (Fig 1) as well as body size data across Triassic-aged *Lystrosaurus* assemblages to investigate whether *Lystrosaurus* from northern Pangea had differing life histories than its southern Pangean relatives.

## Materials and methods

### Geometric morphometric analysis of skull shape

An important question to consider when comparing the bone histology of *Lystrosaurus* specimens from China and other regions is whether the same species occur in the different

geographic areas. Although the first material of *Lystrosaurus* collected in Xinjiang was referred to the South African species *L. murrayi* [55], a total of seven endemic *Lystrosaurus* species also have been described [56–60]. Colbert [33]; also see [61–63], considered some Chinese specimens to represent *L. murrayi* and *L. curvatus*, and others to represent endemic species. More recent authors have been split over whether *Lystrosaurus* specimens from Xinjiang represent exclusively endemic species or a mixture of endemic and more cosmopolitan species, as well as the number of valid endemic species (e.g., [35, 40, 64–67]).

A comprehensive taxonomic revision of *Lystrosaurus* from Xinjiang is beyond the scope of this paper. However, to gain insight into whether Chinese *Lystrosaurus* specimens are likely to be taxonomically distinct from South African specimens, we conducted a preliminary two-dimensional geometric morphometric analysis of skull shape. The skulls were analyzed in three perspectives—anterior, dorsal, and lateral–following the methodology of [68]; also see [67]. The anterior shape analysis included 45 individual specimens, 15 from China and the remaining 30 from South Africa. The dorsal shape analysis included 107 specimens, 14 from China and 93 from South Africa. Lastly, the lateral analysis included 95 specimens, 9 from China and 86 from South Africa. Landmarks were collected using the tpsDig2ws software [69]. In the case of the anterior and dorsal views, landmarks were digitized on only the left half of the skull. The lateral view digitization included a combination of Type I and Type II landmarks (12 total) with a single curve of six semilandmarks along the outer edge of the snout between landmarks number 1 and 11 (Fig 2). The anterior and dorsal views did not contain any semilandmarks, and have 10 and 13 landmarks, respectively. Because taphonomic deformation can complicate the interpretation of shape variation at low taxonomic levels in dicynodonts [70], data collection focused on specimens that appeared to show relatively little evidence of distortion. See the S1 File for additional details of the morphometric analysis.

The coordinate data were processed and analyzed further with the Geomorph R package [71], where they underwent a generalized Procrustes superimposition. Principal components analyses were used to visualize major patterns of shape variation, and a Procrustes ANOVA was run on each dataset to assess the statistical relationship between shape and country of origin using the method of [72] and the Geomorph package in R. A canonical variates analysis was run using the CVA function of the Morpho package [73] in R to further explore the association between shape and geographic location, as well as the accuracy with which specimens could be classified using skull shape (see *Results*).

## Histological sample

Twenty skeletal elements (humerus, femur, tibia, radius, ulna, fibula, and ribs; Table 1; Fig 1) of *Lystrosaurus* sp. were selected for consumptive analysis from the collections of the Institute of Vertebrate Paleontology and Paleoanthropology (IVPP). The majority of the fossils were recovered from strata approximately 30–50 meters above the base of the Jiucaiyuan Formation in the South Taodonggou area, Turpan-Hami Basin, Xinjiang Uygur Autonomous Region, China. One element (IVPP V 27127) came from a nearby locality in Zhaobishan and was recovered about 312m above the base of the Jiucaiyuan Formation [74]. The Jiucaiyuan Formation overlies the Guodikeng Formation lithologically (= upper Wutonggou low-order cycle), and its lower contact is characterized by a facies change from lacustrine to fluvial deposits [74, 75]. This facies change is interpreted as the boundary between the Permian and Triassic [74] and recent work suggests that the Jiucaiyuan Formation is late Induan-early Olenekian in age [74]. As noted above, the lack of clarity regarding Chinese *Lystrosaurus* taxonomy means that we were unable to assign species-level identifications. However, the majority of our histologic samples come from four individuals. Skeletal elements are differentiated by specimen

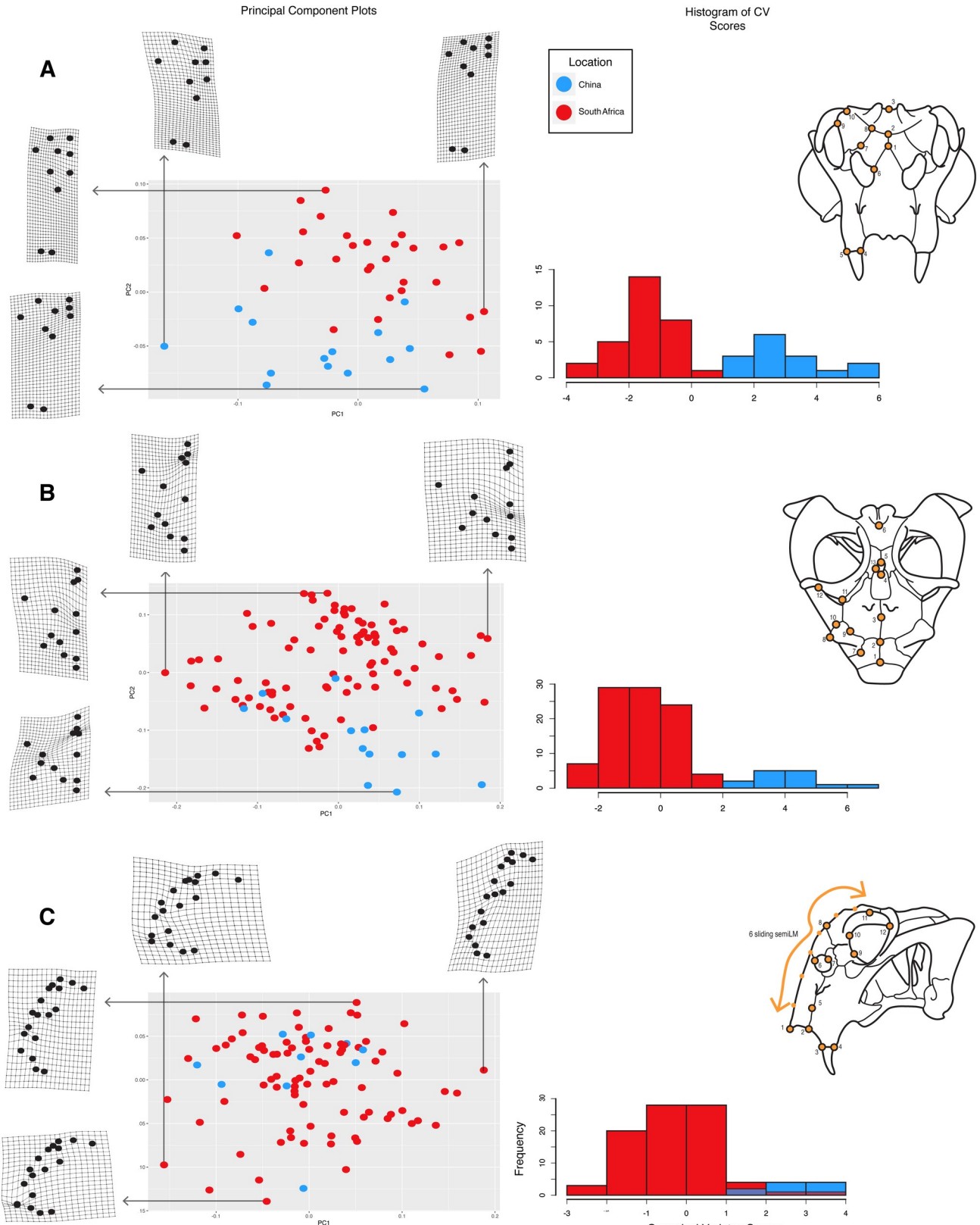

**Fig 2. Results of the morphometric analysis comparing specimens of *Lystrosaurus* from China and South Africa.** Principal components plots and histograms of canonical variates analysis results for the three perspectives (Anterior–A; Dorsal–B; Lateral–C). Skull drawings on the right correspond to

the orientation for each analysis. Orange dots on the specimens show placement of landmarks used in the 2D geometric morphometric analysis (see S1 File for more details). Blue dots represent specimens from China and red dots represent specimens from South Africa. Arrows point to the warp grids showing corresponding deformation along each principal component axis. Histograms show frequency of a recovered canonical variate value and show the distinct differences between Chinese and South African specimens, with overlap only occurring in a few specimens in the lateral orientation.

numbers followed by a letter, in the case of associated individuals (e.g., IVPP V #-a, -b, etc.) The remaining eight specimens are disarticulated elements collected from the same locality and are assigned to one specimen number IVPP V 27124, differentiated by IVPP V 27124–1, -2, etc. In addition to the histologic sample, 50 femora were measured from Early Triassic specimens compiled from the literature [22, 37] and from personal observations to contextualize the range of sizes in this monodominant vertebrate assemblage (S1 Dataset). Additional body size proxies, including basal skull length (S1 Dataset), were measured from *Lystrosaurus* specimens housed in collections at the National Museum (NMQR), the Evolutionary Studies Institute (ESI), the Iziko South African Museum (SAM), the Field Museum of Natural History (FMNH), the American Museum of Natural history (AMNH), and the Burke Museum of Natural History and Culture (UWBM). Cranial and postcranial measurements were made using Neiko digital calipers.

**Table 1. Postcranial element length and maximum midshaft diameter for the twenty skeletal elements of *Lystrosaurus* sp. from the Jiucaiyuan Formation that were histologically sampled.** Asterisk indicates fragmentary or incomplete specimens with accompanying estimates based on complete specimens in this dataset.

| | Spec. No. | Preserved Length (mm) | Estimated Length (mm) | Max Midshaft Diameter (mm) | Growth Marks | Size Class |
|---|---|---|---|---|---|---|
| **Femur** | | | | | | |
| | IVPP V 27124–7 | 122.38 | | 21.43 | 0 | II |
| | IVPP V 27124–6 | 62.74* | 129 | 16.22 | 0 | II |
| | IVPP V 27126a | 62* | 162 | | 0 | II |
| | IVPP V 27125a | 182 | | 28 | 2 | III |
| | IVPP V 26548a | 192 | | 14.09 | 2 | III |
| | IVPP V 27127 | 240* | 255 | 33.5 | 0 | IV |
| **Tibia** | | | | | | |
| | IVPP V 27126b | 92 | | | 0 | II |
| | IVPP V 27124–4 | 56.62* | 94 | 15.86 | 0 | II |
| | IVPP V 27124–3 | 53.26* | 98 | 15.71 | 0 | II |
| | IVPP V 26548b | 148 | | 10.47 | 2 | III |
| | IVPP V 27125b | 148* | 212 | 10.36 | 2 | III |
| **Fibula** | | | | | | |
| | IVPP V 27124–1 | 46.37* | 86 | 9.76 | 0 | II |
| | IVPP V 27126c | 86 | | 5.89 | 0 | II |
| | IVPP V 26548c | 140 | | 9.02 | 2 | III |
| | IVPP V 27125c | 140 | | 15.84 | 2–3 | III |
| **Humerus** | | | | | | |
| | IVPP V 27124–8 | 151.77 | | 23.96 | 1 | III |
| **Radius** | | | | | | |
| | IVPP V 27126d | 62 | | | 0 | II |
| **Ulna** | | | | | | |
| | IVPP V 27126e | 75 | | | 0 | II |
| | IVPP V 27124–2 | 47.32* | 93 | 17.52 | 0 | III |
| **Ribs** | | | | | | |
| | IVPP V 27124–5 | 58.63* | N/A | 48.76–56.33 | EFS? | ? |

## Thin section preparation

Thin sections were made following standard osteohistological techniques [76]. Eight of the specimens were sectioned at the University of Washington (UW); the remaining twelve were sectioned at the Institute of Vertebrate Paleontology and Paleoanthropology (IVPP). At the UW, mid-diaphyses were embedded in Epothin Epoxy/Resin 2, sectioned to a thickness of approximately 2 mm on a high-precision diamond-edged saw (Isomet 1000) and glued to glass slides using Devcon 2-Ton epoxy. Slides were ground on a Metaserv 3000 lapidary plate until the specimen was 80 µm thick or until optical clarity was reached. At the IVPP, midshafts were cut and embedded in polyester resin, sectioned into blocks using an Exact 300CP band saw, and glued to glass slides using Technovit 4000 resin. Slides were ground to optical clarity using an automated lapidary plate (Exakt 400CS). High magnification and composite images were taken using a Nikon Eclipse LV100POL microscope under plain and cross-polarized light with a lambda filter. Composite images were processed using Nikon NIS-Elements AR (version 5.20.02) imaging software. High-resolution images are available at Morphobank under project number P4023.

Ontogenetic maturity was assessed through bone tissue composition, articular surface texture and morphology, and size compared to other specimens of *Lystrosaurus* from the Jiucaiyuan Formation. The largest femur, estimated at 255 mm, is of comparable length to the largest known femur (*L. declivis* SAM-PK-K8038) from the Triassic of South Africa, which is 202 mm long. Therefore, we follow the size classes established by Botha [22] for southern Pangean *Lystrosaurus*, but note that our sample lacks Size Class I, or elements less than 40% maximum known size. The remainder of the elements were split into three size categories for analysis (Table 1; Fig 1). Size Class II includes the smallest and presumably most immature specimens in the current sample, with femoral lengths that are approximately 120–160 mm long (47–63% of the maximum femur length observed), tibiae and fibulae that are less than 100 mm long, and associated forelimb elements. All but the proximal tibia fragment (IVPP V 27124–3) have poorly formed joint surfaces and show pitting indicative of epiphyseal cartilage attachment in immature individuals [77]. By comparison, the joint surfaces of elements in the larger size classes are more robustly built and more completely ossified. The relative degree of epiphyseal ossification combined with histological features caused us to deviate slightly from Botha's [22] size classes by grouping specimens that are 40–65% maximum known size as Size Class II, rather than her 60% MKS cutoff. Our reasoning for this slight change is that the bone tissue is consistently and characteristically immature (i.e., highly vascularized plexiform to radial tissue without LAGs) in all of the associated elements that are 63% MKS. However, the incomplete tibia (IVPP V 27124–3, estimated length of 94 mm) that is estimated to be within Size Class II has a more developed and ossified joint surface compared to similarly sized tibiae (IVPP V 27124–4, 94 mm in length) so it is likely that with more sampling, the split between Size Class II and III will be refined. Size Class III includes immature specimens that are approximately 70% maximum known size (again, based on femoral lengths from larger, complete, Jiucaiyuan specimens). Finally, Size Class IV includes a single femur with an estimated length of 255 mm (the specimen measures 240 mm and is missing the proximal shaft and head of the femur). To our knowledge, it is the largest femur of *Lystrosaurus* yet recovered from the Jiucaiyuan Formation, and is ~ 5 cm longer than the largest femur (discussed above) recovered from the Triassic of South Africa. Therefore, it presumably represents the most skeletally mature individual in the current sample. However, histological analysis will provide critical context for the relative maturity of this, and all other specimens sampled [78].

## Bone tissue terminology

Bone tissue textures seen in *Lystrosaurus* and other therapsids vary widely from highly vascularized and disorganized woven-fibered bone to organized vascular canals in circumferential

layers of parallel-fibered to lamellar bone [19, 30, 79–83]. The overall rate of bone growth affects the degree of organization of the extracellular matrix, with mineral fibers in a lamellar organization under slower growth, and disorganized woven-fibered textures under fast growth. Parallel-fibered bone is an intermediate tissue-type but it frequently occurs in the context of woven-fibered and lamellar bone. When large portions of the cortex consist of parallel-fibered bone, the mineralized extracellular matrix has an alternating anisotropic pattern that looks streaky under cross-polarized light. In many cases, the portion of woven or parallel-fibered bone that sits between vascular canal spaces is so small that birefringent properties are difficult to discern. When this is the case, it is possible to use the morphology and density of osteocyte lacunae (spaces where bone cells sit in living bone) to approximate the relative organization of mineralized fibers [84, 85]. Woven-fibered bone is statically-derived and includes large, globular, and often densely packed osteocyte lacunae within the mineralized matrix [84, 86]. In contrast, lamellar and parallel-fibered bone are dynamically-derived, and have lenticular or spindle-shaped osteocyte lacunae that are evenly distributed and parallel to the layers of the mineralized matrix [84, 86, 87]. Importantly, in long bone cortices, the fiber network of parallel-fibered bone can either be aligned to the bone's long-axis (i.e., longitudinal fibers) or perpendicular to it (i.e., circular fibers) which can cause differences in lacunar morphology and fiber birefringence in polarized light depending on the plane of section [44]. When parallel fibers are perpendicular to the sectioned plane, they show mass isotropy or monorefringence, similar to woven-fibered bone [85].

Non-lamellar tissues (parallel- and woven-fibered bone) frequently incorporate large vascular spaces that later become infilled with concentric lamellae of parallel-fibered to lamellar bone as primary osteons. The resulting tissue is a fibrolamellar bone complex (sensu Francillon-Vieillot [45]) wherein a disorganized, nonlamellar matrix incorporates a network of centripetally infilled primary osteons. Ricqlès [88] coined much of the terminology used by paleohistologists and he used a restricted definition of the fibrolamellar complex that applied only to bone tissue that strictly had woven-fibered matrix, rather than the combination of woven- and parallel-fibered bone used in other applications of the term. More recently, Stein and Prondvai [85] reinterpreted cortical bone structures using polarized-light microscopy of sauropod thin sections cut from multiple planes. They concluded that the non-osteonal portion of cortical bone was largely parallel-fibered rather than woven and proposed new terminology for what they interpreted as highly organized primary bone (HOPB). They also proposed an additional term, the woven-parallel complex, to combine the organized portions of HOPB with the more traditional terminology used in bone development and growth (for more, see [89]). The term woven-parallel complex describes a broad range of bone tissues with nonlamellar matrices and the term fibrolamellar complex is maintained as a subset of the woven-parallel complex where the cortical bone is densely vascularized with abundant woven-fibered bone forming the scaffold between vascular canals which is then surrounded by parallel-fibered or lamellar bone [44, 89]. This new terminology recognizes the developmental aspects of osteogenesis and the accumulation of osteoblasts within mineralizing osteoid. However, it may unnecessarily limit the role of woven-fibered bone to a transient tissue type typical of juvenile bone. Woven-fibered bone is known to be present in the typical development, maintenance, and repair of bone tissue throughout ontogeny in vertebrate groups and its abundance in cortical bone may depend on physiological processes rather than development alone [87]. In our histological descriptions below, we follow the recommendation [44, 89] to include lacunar morphology to estimate the proportion of woven- and parallel-fibered bone within the fibrolamellar and woven-parallel complexes. We also acknowledge the histovariability of parallel- and woven-fibered bone that can occur simultaneously and often in different proportions within the same section, forming a spectrum of tissue types that can be difficult to define.

## Results

### Geometric morphometric analysis of skull shape

Anterior view: The first five principal components capture 81.41% of the total shape variance in anterior view (PC1–30.57%; PC2–22.03%; PC3–13.82%; PC4–8.71%; PC5–6.26%) (Fig 2A). Shape is a statistically significant predictor of geographic location for the anterior view dataset (p = 0.001), and the canonical variates analysis correctly assigned geographic location (i.e., China or South Africa) 100% of the time.

PC1 primarily describes relative proportions of the snout. The subnarial region of the snout and caniniform process are dorsoventrally shorter relative to the height of the nasals and the anterior orbital margin in specimens with low PC1 scores, and there is greater lateral projection of the prefrontals relative to the caniniform process (Fig 2A). In contrast, the subnarial region and the caniniform process are proportionally much deeper relative to the anterior orbital margin and nasals in specimens with high PC1 scores, and the prefrontals and caniniform process are more closely aligned along a vertical axis. PC2 also captures aspects of skull widening and deepening, although in this case the differences are concentrated in the region of the nasals and prefrontals. Specifically, the prefrontal-nasal region is dorsoventrally shorter in specimens with low PC2 scores, and the dorsal and ventral extrema of the prefrontal are displaced laterally relative to the external naris and the lateral end of the nasofrontal suture. Specimens with high PC2 scores have dorsoventrally taller prefrontal-nasal regions, and the dorsal and ventral extrema of the prefrontal are more closely aligned with the external naris and the lateral edge of the nasofrontal suture. The Chinese and South African specimens are most clearly differentiated along PC2, with the Chinese specimens generally having lower PC2 scores.

Dorsal view: The first five principal components capture 79.75% of the shape variance in dorsal view (PC1–26.77%; PC2–23.57%; PC3–17.95%; PC4–6.18%; PC5–5.26%) (Fig 2B). Shape is a statistically significant predictor of geographic location for the dorsal view dataset (p = 0.001), and the canonical variates analysis correctly assigned specimen location 100% of the time.

Shape variation along PC1 is dominated by differences in the relative lengths of the interorbital region and the pineal foramen, and by the degree of lateral projection of the orbital rim. Specimens with low PC1 scores are characterized by an anteroposteriorly longer interorbital region and a shorter pineal foramen, and an orbital rim that does not project strongly laterally (Fig 2). The pineal foramen is relatively longer anteroposteriorly in specimens with high PC1 scores, and the anterior and posterior corners of the orbital rim (formed by the prefrontal and postorbital bar, respectively) project laterally away from the skull roof. PC2 also captures shape variation associated with relative proportions of the skull roof, as well as orbit shape. Specimens with low PC2 scores have an anteroposteriorly shorter pineal foramen and longer frontals, and the orbit is anteroposteriorly shorter, with anterior and posterior corners that are relatively aligned. The pineal foramen and orbit are relatively longer in specimens with high PC2 scores, and the posterior corner of the orbit (formed by the postorbital bar) projects farther laterally than the anterior orbital margin (formed by the prefrontal). As with the anterior view dataset, the Chinese and South African specimens are most clearly differentiated along PC2, with the Chinese specimens generally having lower PC2 scores.

Lateral view: The first five principal components capture 77.23% of the shape variance in lateral view (PC1–32.64%; PC2–18.57%; PC3–11.66%; PC4–8.58%; PC5–5.76%) (Fig 2C). Shape is a statistically significant predictor of geographic location (p = 0.001), and the canonical variates analysis had an overall classification accuracy of 95.78%, misclassifying two Chinese specimens in the dataset.

PC1 describes differences in the profile of the snout, position of the caniniform process relative to the orbit, and the anteroposterior dimensions of the orbit (Fig 2C). Specimens with low PC1 scores have a lower, more anteriorly-convex snout profile with prefrontals that do not project much above the level of the skull roof, and the ventral edge of the caniniform process is located below the anterior orbital margin. The anteroposterior length of the orbit is relatively long. In contrast, specimens with high PC1 scores, have much taller, flatter snout profiles and more dorsally-projecting prefrontals, as well as anteroposteriorly shorter orbits and caniniform processes that are located anterior to the anterior orbital margin. PC2 also captures aspects of snout profile, including the degree of anterior projection of anteroventral corner of the premaxilla and the length of the caniniform process, as well as the dorsoventral height of the orbit. The tip of the premaxilla projects more anteriorly in specimens with low PC2 scores, the caniniform processes are relatively short, and the orbits are dorsoventrally shorter. Specimens with high PC2 scores have taller orbits, flatter snout profiles, and longer caniniform processes. There is little differentiation of the Chinese and South African specimens on the first three PC axes.

Taken together, our morphometric analyses suggest that there are consistent differences in skull shape between *Lystrosaurus* specimens from China and from South Africa, particularly in anterior and dorsal view. Based on these results, we tentatively suggest that the Chinese and South African populations were distinct at the species level. However, because our analyses do not directly address potential synonymies among the named Chinese species, and the fact that not all of the specimens we sampled histologically are associated with cranial material, we conservatively refer all of our histological samples to *Lystrosaurus* sp.

## Osteohistology overview

*Lystrosaurus* limb elements, regardless of size, have characteristically thick cortices. Cortical thickness ($K$) measures the proportional diameter of the medullary region relative to the total cross-sectional diameter and remained nearly consistent at 0.63 and 0.64 in the smallest and largest femora in our dataset, respectively (see S1 File). Medullary cavities are typically infilled with trabecular bone that gradually transitions to a compact cortex. Similar to the bone histology of other dicynodonts and therapsids more broadly, the predominant bone tissue in all sampled size classes of *Lystrosaurus* is well-vascularized cortical bone made up of varying proportions of woven- and parallel-fibered bone with laminar, plexiform, and longitudinally-oriented primary osteons, indicative of rapid osteogenesis [19, 30, 79, 80–83]. Longitudinal canals that frequently anastomose are arranged in circumferential layers in Size Class II whereas laminar vascular canal orientations are common in Size Class III. In Size Class III and IV, very little of the cortex is remodeled and when secondary osteons are present, they are restricted to the deepest portion of the cortex.

In the small individuals of *Lystrosaurus* studied here, rapidly deposited tissue is interrupted by either LAGs or annuli. These growth marks are found in multiple regions of the cortex, including deep in the cortex and along the sub-periosteal margin. In some individuals, there is variability in the type of growth mark that is deposited among the skeletal elements sampled. For example, the first (i.e., osteologically deeper) growth mark of IVPP V 26548 varies from a LAG in the femur and tibia, to an annulus in the fibula. Interestingly, the largest and presumably most skeletally mature femur that was histologically sampled, shows uninterrupted cortical growth, suggesting that *Lystrosaurus* from the Jiucaiyuan Formation had a high intrinsic rate of growth that could periodically arrest.

Our results are consistent with a recent report of *Lystrosaurus* bone histology from the same formation with respect to the preponderance of highly vascularized, rapidly deposited

cortical bone [40]. However, contrary to Han et al. [40], we do not find evidence of peripherally slowed growth rates in individuals that are as large (~60% MKS) and larger than the maximum size sampled in their study. Additionally, we propose an alternative interpretation to the assignment of LAGs in the laminar to plexiform tissue reported in ([40]: Figs 2G, 2H, 3C, 5, 6D, 6G, 6H) as bright lines that appear in the context of periosteal bone formation ([90]: Fig 3C; [91]: Fig 3). Finally, we report fewer instances of growth marks, particularly in the smallest size class where no evidence of episodic growth marks is seen (contra [40]). When present, we can confidently trace one to three growth marks (as either LAGs or annuli) around the circumference of individuals that are approximately 70% maximum known size.

## Size Class II

The smallest association of elements includes cranial and postcranial material (hind limb and forelimb elements) that was collected in situ. A femur, tibia, fibula, ulna, and radius were selected for consumptive sampling (composite images available on Morphobank under project number P4023). The femur measures 122.38 mm in length. When compared to femora from larger, presumably more mature individuals, this femur (IVPP V 27126a) is approximately 50% maximum known size. The bone tissue in each of the five associated limb elements is extremely well-vascularized with longitudinally-oriented primary osteons arranged in circumferential layers with some radial anastomoses (Fig 3). Woven fibered bone matrix and large, densely distributed osteocyte lacunae surround the primary vascular canals (Fig 3B and 3E). Vascular canal size and density remain consistently high from the endosteal to the periosteal margin indicating that this individual grew rapidly until death (Fig 3D and 3E). There is no evidence of episodic growth cycles (as either LAGs or annuli) in this individual.

A right proximal fibula (IVPP V 27124–1), right proximal tibia (IVPP V 27124–3), distal tibia (IVPP V 27124–4), distal femur (IVPP V 27124–6) and complete femur (IVPP V 27124–7) were collected individually but are analyzed together here as immature individuals because limb dimensions and histological features are indicative of sustained, active bone growth at death. The complete femur is approximately 50% maximum known size, similar to IVPP V 27126a. In thin section, each of these disassociated elements has a large medullary cavity infilled with a loose trabecular network. Cortical tissue consists of woven-fibered bone with longitudinally oriented primary osteons loosely arranged in circumferential rows. In the proximal tibia, vascular canal size decreases substantially in a regionalized band deep to the subperiosteal edge but it rebounds to the typical highly vascularized tissue more peripherally, likely representing a temporary shift in growth rate (Fig 3F and 3G). In the fibula, there are regional differences in vascular canal size and organization that likely represent functional differences in bone apposition rates [92]. For example, longitudinal vascular canals are organized into stacked, circumferential layers on the medial side of the proximal fibula (IVPP V 27124–1) (Fig 3B and 3C) whereas the lateral side is less organized (Fig 3A). The amount of densely packed osteocyte lacunae in woven-fibered bone is higher in the less organized areas, but many large, globular lacunae are still apparent between primary osteons and in thin sheets in the more organized regions (Fig 3B and 3C). In more organized areas of the highly vascularized tissue, the mineralized matrix surrounding the vascular canals should not be confused with lines of arrested growth (Fig 3B and 3C). At low magnifications, delineated bands of extracellular matrix separate the primary vascular space in periosteally accreted bone [91]. However, at high magnifications, these accretion lines, or bright lines (sensu [91]), do not form continuous rings around the cortex and should not be confused as growth marks or as slower forming tissues like parallel-fibered or lamellar bone [45]. Bright lines lack osteocyte lacunae and represent the saltatory activity of the periosteum when new bone is accreted [45,

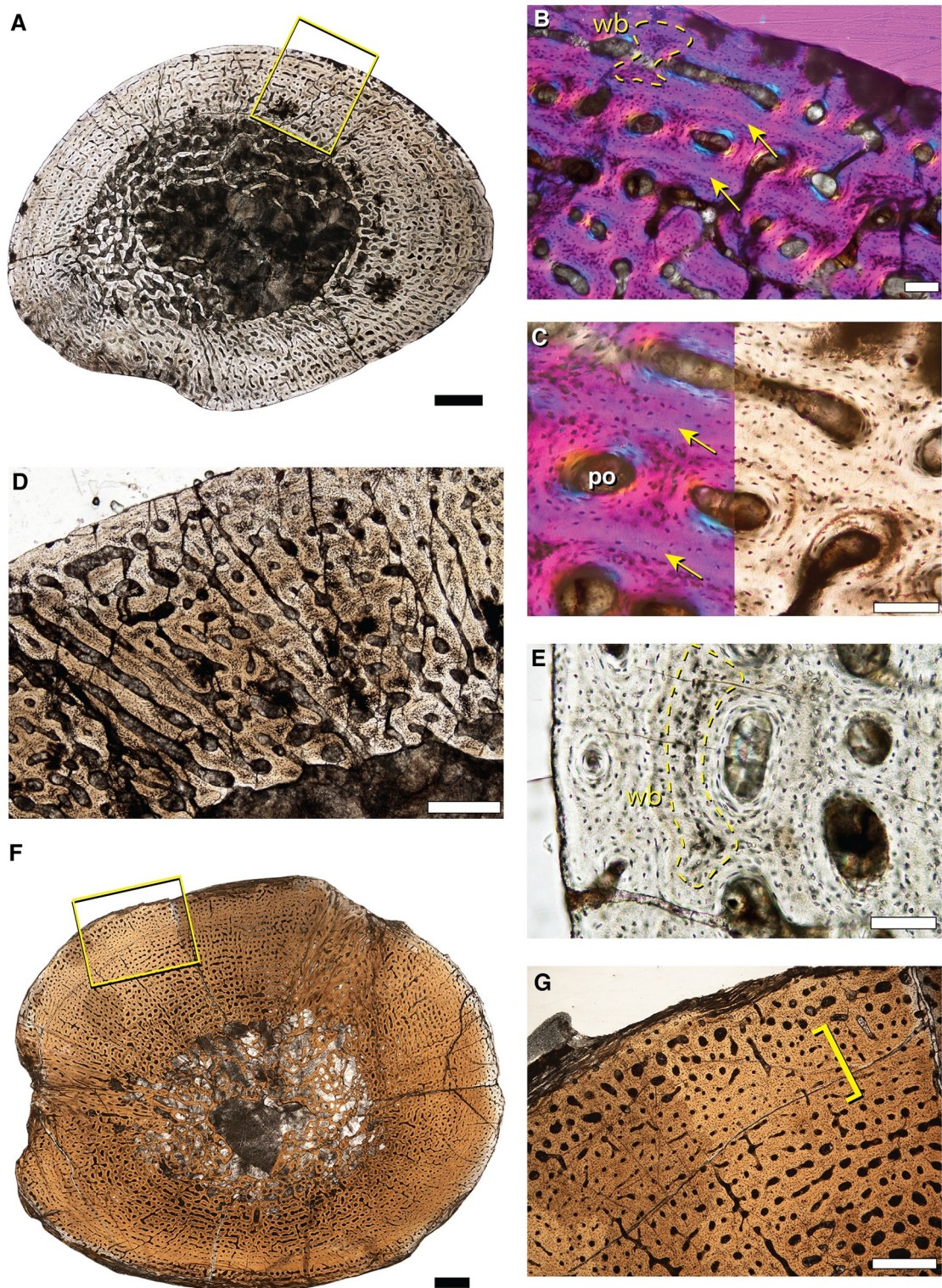

**Fig 3. Representative bone histology of Size Class II from the mid-shafts of *Lystrosaurus* sp. hind limb elements. A**, fibula (IVPP V 27124–1) consisting of well-vascularized tissue with longitudinally-oriented primary osteons in circumferential layers. **B**, 10X magnification of yellow square in **A**, arrows indicate bright lines in the extracellular matrix of periosteally accreted bone and should not be confused as growth marks. **C**, 20X magnification of **B**. **D**, fibula (IVPP V 27126c) cross section consisting of longitudinally and radially oriented primary osteons and dense osteocyte lacunae. **E**, 20X magnification of fibula (IVPP V 27126c)

shows woven-fibered bone (outlined in yellow) in the peripheral cortex indicating rapid bone deposition at death. **F**, cross section of a proximal tibia (IVPP V 27124–3) with a slight decrease in vascular canal sizes towards the sub-periosteal edge, magnified in **G**. wb = woven-fibered bone; po = primary osteon. Scale bars = 1 mm in **A** & **F**, 500 μm in **D** & **G**, 100 μm in **B**, **C** & **E**.

91]. In specimens in Size Class II, highly vascularized longitudinal canals in woven- to parallel-fibered matrix continue to the sub-periosteal edge, indicating that individuals died while actively growing.

## Size Class III

A femur (IVPP V 27125a), tibia (IVPP V 27125b), and fibula (IVPP V 27125c) are associated as one individual that is approximately 70% maximum known size based on femoral length. Unlike smaller sampled elements, the cortex is interrupted by two growth marks (Fig 4). In these associated elements, a deep cortical LAG is sandwiched between large vascular canals towards the endosteal margin and smaller vascular canals more peripherally (Fig 4A and 4B, lowermost arrows in F, K). In the fibula, the LAG is superficial to a region of immature secondary osteons that sit in the deep cortex (Fig 4J). The second of the two growth marks is more variable, in part due to the fragmentary nature of the subperiosteal edge of the femur. In the femur, a poorly vascularized annulus of parallel-fibered to lamellar tissue sits along the sub-periosteal edge and is more apparent in thin areas of the cortex (Fig 4D). By contrast, thicker regions of the outer cortex show a mixture of longitudinal and circular primary osteons in circumferential layers that approaches a laminar organization and continues to the incomplete sub-periosteal edge (Fig 4C). In the tibia and fibula, the outer cortical growth mark is made up of two LAGs in the tibia, and an annulus of avascular parallel-fibered tissue in the fibula (Fig 4E, 4J and 4K). Interestingly, the mineral preservation of the tibia and fibula are such that the mineralized bone matrix is clearly visible between each circumferential layer of vascular canals (Fig 4F, 4K and 4L). These thin, mineralized lines of extracellular matrix should not be confused for growth marks and instead are the mineralizing front of periosteally accreted bone [45, 90]. The vascular orientation shifts in the hind limb elements of this individual from reticular to laminar in the femur (Fig 4B), and from disorganized longitudinally oriented primary osteons to circumferentially organized longitudinal primary osteons in the tibia and fibula (Fig 4F and 4I). In addition, primary osteon size decreases towards the outermost periosteal margin but vascular canals remain open to the sub-periosteal edge (Fig 4F and 4G). The only exception to this pattern is in the thinnest region of the cortices, particularly in the fibula, where the sub-periosteal edge is avascular and consists of well organized, parallel-fibered bone (Fig 4D, 4I and 4K). However, this organized tissue is discontinuous around the cross section, suggesting that bone growth slowed at differential rates in regions of the outer cortex, but did not fully arrest.

A slightly larger specimen (IVPP V 26548) is an association of hind limb and forelimb elements as well as at least one rib that was previously histologically sampled by Han et al. 2021 [40]. The femur (IVPP V 26548a), tibia (IVPP V 26548b), and fibula (IVPP V 26548c) show plexiform tissue interrupted by two growth marks, with some variation in the outer growth mark, especially in the fibula where it splits into two LAGs (Fig 5). In the femur and tibia, the deeper growth mark is distinguishable as a dramatic reduction in vascular canal size, immediately followed by a return to the typical, large vascular canals in the middle of the cortex (Fig 5A, 5B and 5D). Throughout the middle cortex of the femur, the deeper growth mark changes in thickness. In thinner regions of the cortex, the growth mark appears as a cross-cutting LAG separating smaller vascular canals from larger canals more superficially. In the thicker regions

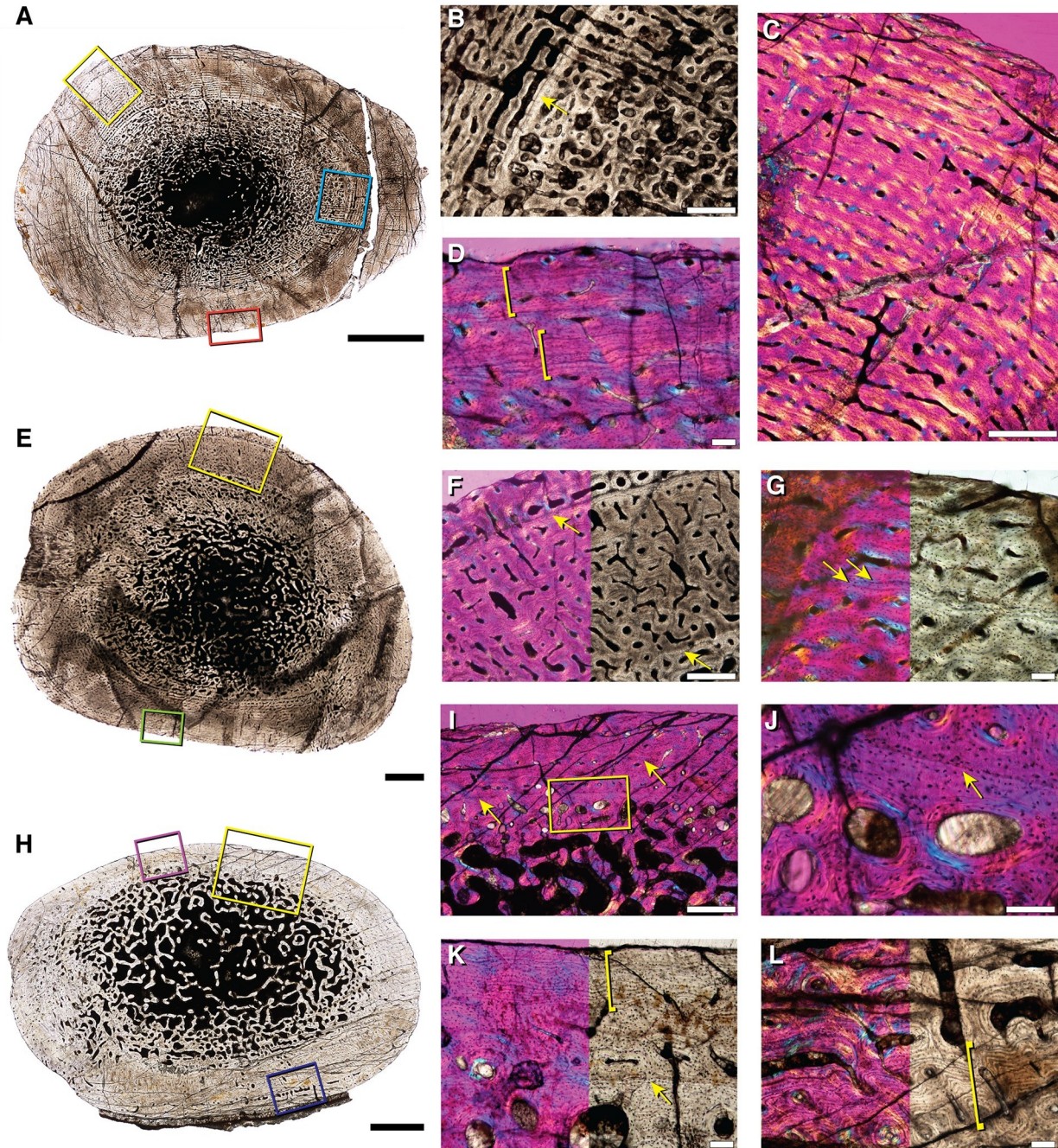

**Fig 4. Bone histology of associated hind limb elements of *Lystrosaurus* sp. representative of Size Class III. A**, femur (IVPP V 27125a) with a deep cortical LAG in **B** (blue box in **A**). The outer cortex has regions of high vascularity seen in **C** (yellow box in **A**), and regions marked by a nearly continuous peripheral growth mark of parallel-fibered tissue (denoted by yellow brackets) in **D** (orange box in **A**). **E**, tibia (IVPP V 27125b) has two cortical growth marks seen in **F** (yellow box in **E**; yellow arrows in **F**). The outer growth mark can variably be seen as a double LAG in **G** (green box in **E**; yellow arrows in **G**). **H**, the fibula (IVPP V 27125c) has a thinner cortex interrupted by two LAGs seen in **I** (yellow box in **H**; yellow arrows in **I**). **J**, magnification of yellow box in **I** shows the cross-cutting inner cortical LAG (arrow in **J**). **K**, thinner regions of the cortex appear avascular (purple box in **H**), with stacked mineralized lamellae of parallel-fibered bone (yellow bracket), whereas thicker regions (dark blue box in **H**) show peripheral pulses of vascularized growth, within the lamellated tissue, seen in **L** (yellow bracket). Scale bars = 5 mm in **A**, 1 mm in **E**, 2 mm in **H**, 500 μm in **B**, **C**, **F** & **I**, and 100 μm in **D**, **G**, **J**—**L**.

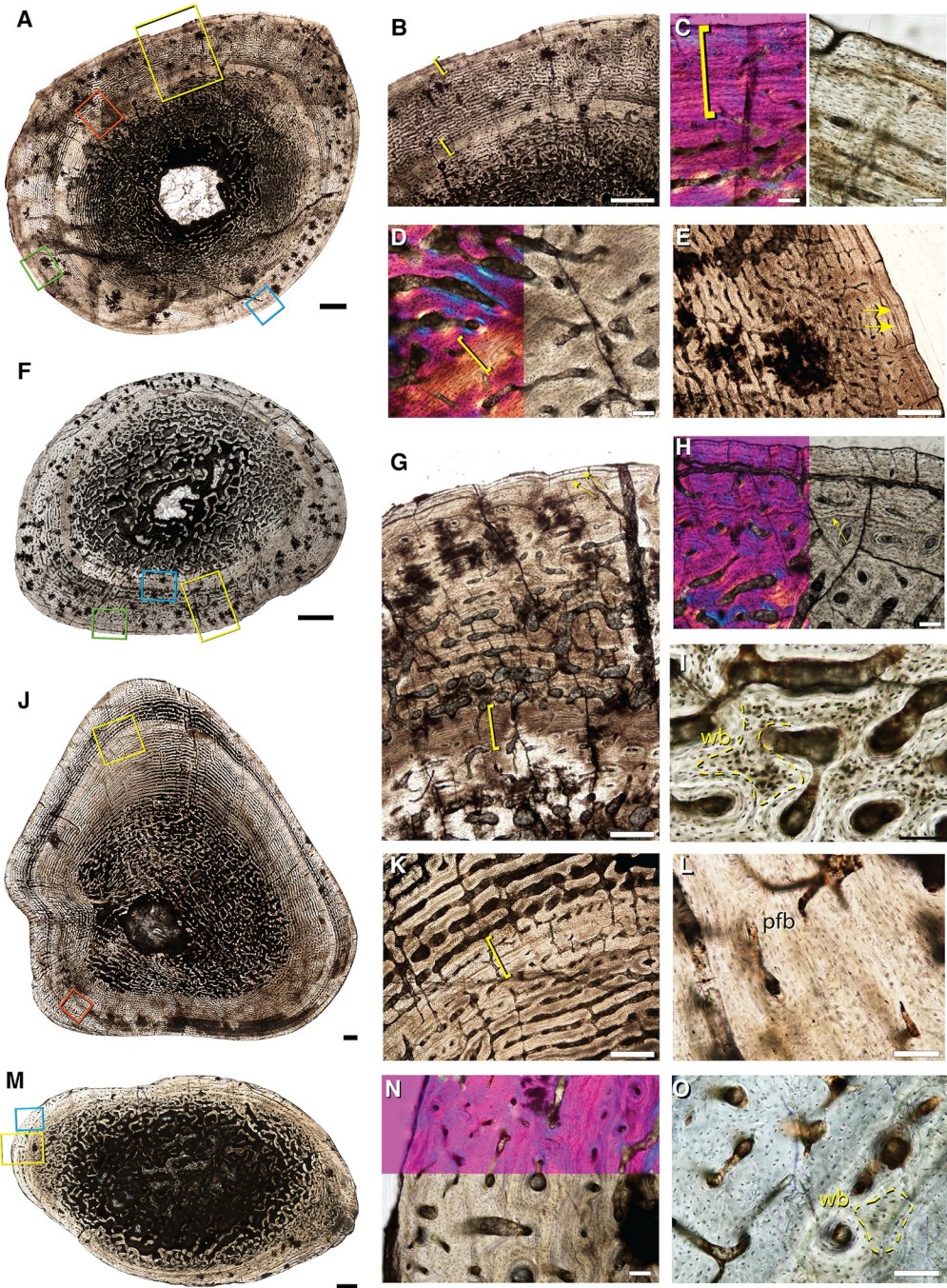

**Fig 5. Bone histology of associated elements of *Lystrosaurus* sp. representative of Size Class III. A**, femur (IVPP V 26548a) has two growth marks. **B**, a deep, mid-cortical annulus (yellow box in **A**; lower yellow bracket in **B**), avascular lamellar tissue makes up the second growth mark (upper yellow bracket in **B**). **C**, magnification of peripheral lamellar tissue, left panel reflects blue box in **A**, right panel reflects green box in **A**. **D**, first (osteologically deeper) annulus (yellow bracket) in plane and cross-polarized light with a lambda filter. **E**, magnification of blue box in **A** of outer cortical growth mark that varies in thickness around the periphery of the thin section, here consisting of two LAGs (yellow arrows in **E**). **F**, associated fibula (IVPP V 26548c) highlights a similar arrangement of deep and more superficial growth marks consisting of, at times, numerous layers of lamellar tissue in an annulus. **G**, magnification of yellow box in **F** showing two growth marks; annulus (yellow bracket) and peripheral LAGs (yellow arrows). **H**, magnification of green box in **F** showing variation in the outer growth mark such that in some regions it appears as a thicker region of lamellar tissue with one cross-cutting LAG. **I**, high magnification image of the associated tibia (IVPP V 26548b) showing the presence of woven-fibered bone between primary osteons within the middle cortex. **J**, humerus

with mid-cortical annulus of reduced vascularity. **K**, magnification of yellow box in **J** of annulus (yellow bracket). **L**, high magnification of parallel-fibered bone making up the annulus in the humerus (orange box in J). **M**, ulna (IVPP V 27124–2) with a thinner overall cortex made up of parallel-fibered and small amounts of woven-fibered bone. **N**, magnification of yellow box in **M** highlighting the thin cortex in plane and cross-polarized light with a lambda filter. **O**, high magnification of outer cortex of the ulna showing small areas of woven-fibered bone (blue box in **M**). wb = woven-fibered bone; pfb = parallel-fibered bone. Scale bars = 1 mm in **A**, **B**, **F**, **J**, & **M**, 500 μm in **E**, **G** & **K**, and 100 μm in **C**, **D**, **H**, **I**, **L**, **N**, & **O**.

of the cortex, the growth mark appears as an annulus, or region of little to no vasculature that is not accompanied by a hyper-mineralized line (Fig 5D). In both the femur and tibia, vascular canals reduce in size again towards the sub-periosteal margin, making up the second growth mark of an annulus of avascular parallel-fibered to lamellar tissue (Fig 5C). The fibula shows a mid-cortical annulus that is made up of avascular parallel-fibered to lamellar tissue (Fig 5G). Similar to the femur and tibia, there is a sub-periosteal reduction in vascular canal size with at least two LAGs that can be traced around the entire cortex of the fibula, indicating variation in the type of growth mark deposited in hind limb elements of the same individual (Fig 5E, 5G and 5H). Small, simple vascular canals sit within this peripheral lamellar tissue, which is atypical for an external fundamental system (EFS) indicative of somatic maturity, making it more likely that this lamellar tissue represents a temporary cessation in growth, similar to the annulus recorded in the deeper cortex.

A complete humerus (IVPP V 27124–8) of comparable size to IVPP V 26548 [40], has an exceedingly thick cortex with one mid-cortical growth mark (Fig 5J). The medullary cavity is nearly infilled by a dense network of trabecular bone. The cortex increases in thickness in the anterodorsal region (up in Fig 5J) and consists of highly vascularized laminar bone in woven- and parallel-fibered bone matrix. Vascular canals have a circular orientation and show a marked decrease in size coincident with a mid-cortical LAG (Fig 5K). The LAG is traceable around the entire cross-section and is followed by a pulse of highly vascularized tissue with large canal spaces (Fig 5K). Vascular canal size reduces slightly towards the sub-periosteal margin which could indicate the beginning of another growth mark, similar to the mid-cortical growth mark.

A distal fragment of an ulna (IVPP V 27124–2; original complete length estimated to be 93 mm) has a thin cortex with a large medullary cavity infilled with a loose network of trabecular bone (Fig 5M). The transition from trabeculae to compact cortical bone includes a region of enlarged resorption cavities and immature secondary osteons. The middle and outer cortex is composed of small longitudinal canals that sit in a parallel-fibered matrix where small regions of woven-fibered bone persist (Fig 5N and 5O). Along the sub-periosteal margin, vascular canals are reduced but occasionally open to the bone edge, suggesting that this individual died while still growing (Fig 5M). The proportion of parallel-fibered bone in this element is higher than in hind limb elements, however, there is no evidence to suggest somatic maturity or attainment of maximum size.

## Size Class IV

Unlike the middle size class, the largest sampled femur (IVPP V 27127) does not have LAGs or annuli (Fig 6A). However, much of the cortex is cracked or incompletely preserved, and a large crack is present in the middle cortex, along what might have been a LAG. Despite these cracks, the bone tissue architecture is easily discernible and consists of laminar canals in woven- and parallel-fibered matrix based on the lacunar morphologies at high magnification (Fig 6B and 6C). Woven-fibered bone makes up more of the extracellular matrix in the deep and middle cortex whereas the outer cortex consists of parallel-fibered bone with small areas

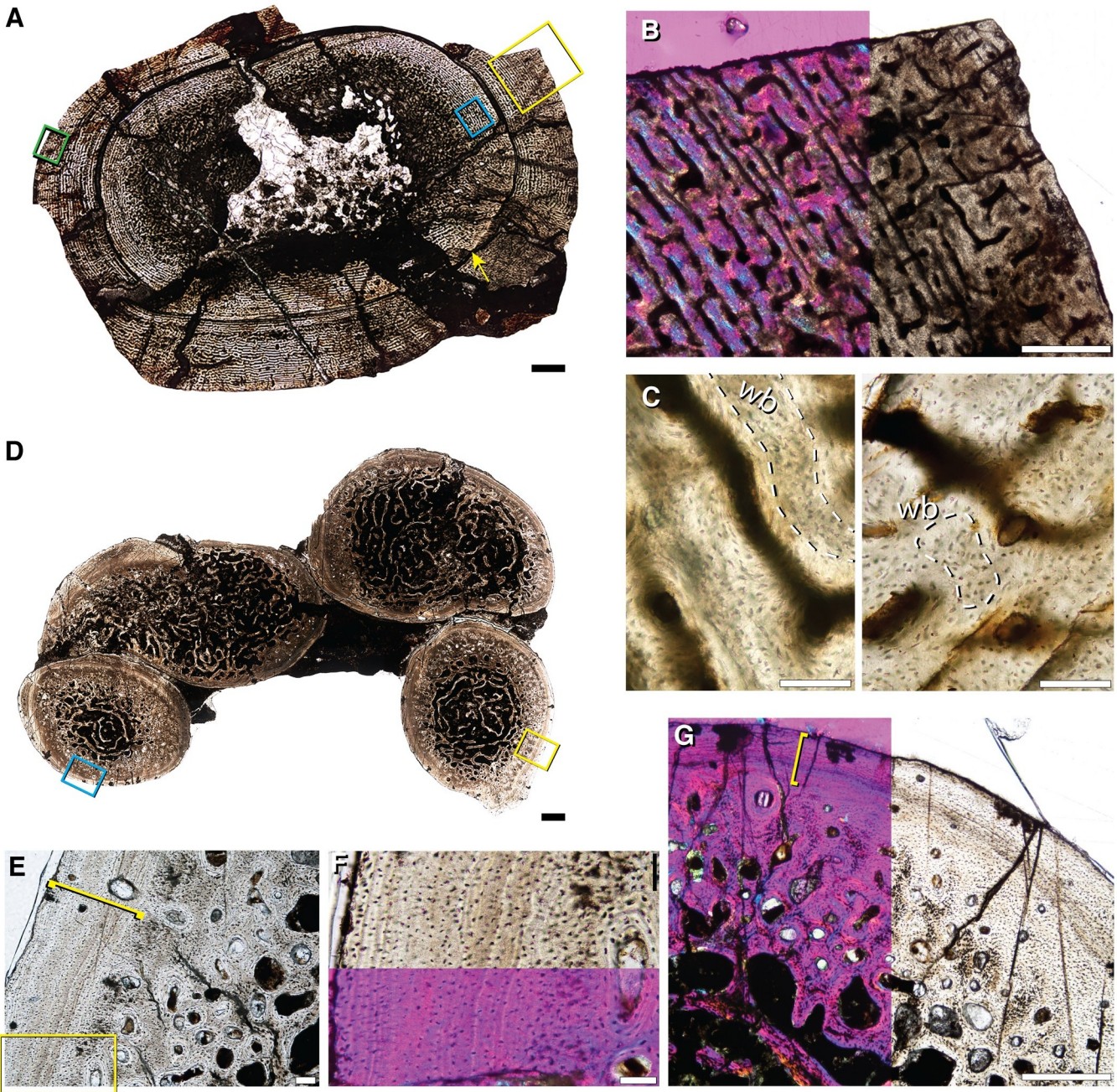

**Fig 6. Bone histology of the largest femur (IVPP V 27127) of *Lystrosaurus* sp. that represents Size Class IV and rib fragments of unknown size. A**, the femur is cracked (yellow arrow is a large circular crack) but preserves regions of the entire cortex (yellow box). **B**, magnification of yellow box in **A** using cross-polarized light with a lambda filter shows well-vascularized, plexiform tissue along the sub-periosteal margin. **C**, left panel shows woven-fibered bone in the deep-middle cortex (blue box in **A**) and outer cortex, right panel (green box in **A**). **D**, rib fragments (IVPP V 27124–5) have infilled medullary cavities, secondary osteons within the deep cortex and a largely avascular compact outer cortex consisting of organized parallel-fibered tissue. **E**, magnification of yellow box in **D** showing the peripheral parallel-fibered tissue in the yellow bracket. **F**, magnification of the yellow box in **E** under plane and cross-polarized light with a lambda filter. **G**, magnification of blue box in **D** similarly shows that peripheral parallel-fibered tissue is present (yellow bracket). wb = woven-fibered bone. Scale bars = 10 mm in **A**, 1 mm in **D**, 500 μm in **B** & **G**, 100 μm in **C**, **E** & **F**.

of woven-fibered bone (Fig 6C). There is no indication of a transition to slowed bone deposition in the small regions of sub-periosteal bone that remain intact, suggesting a sustained rate of growth at death (Fig 6B).

### Unknown size class

Four partial ribs (IVPP V 27124–5) belonging to an individual of unknown size class are preserved in a block of sediment (Fig 6D). In thin section, they have large, infilled medullary cavities that grade into thin, compact cortices. Scarce vascular canals and immature secondary osteons sit in the deeper cortex whereas the outer cortex is avascular and consists of parallel-fibered tissue with well-organized osteocyte lacunae (Fig 6E and 6F). Due to the fragmentary nature of this specimen and the fact that it is not associated with limb bones, it is unclear whether the peripheral parallel-fibered tissue is indicative of an EFS or a temporary cessation in growth. Additionally, ribs grow at different rates than limbs and do not preserve the same record of life history information compared to a midshaft hind limb thin section. Another interesting feature that these ribs reveal is that the cortex differs in thickness between each rib, casting doubt on Ray et al.'s [54] hypothesis that *Lystrosaurus* had exceptionally thick ribs, which they suggested was indicative of an aquatic lifestyle. From this sample, it is clear that *Lystrosaurus* rib cortical thickness depends on where the thin section was taken.

## Discussion

All *Lystrosaurus* individuals sampled from the Jiucaiyuan Formation show abundant, highly vascularized primary bone tissue throughout the majority of the cortices of the hind limb, humerus, and radius. The ulna and ribs showed fewer vascular canals, overall thinner cortices (contra [54]), and more organized primary bone tissue. Three size classes were identified and histologically described as: II) a highly vascularized cortex of circumferential layers of radial and longitudinal vascular canals in ~50% maximum known size individuals: III) highly vascularized plexiform tissue interrupted by one to two growth marks that range from one or two closely spaced LAGs or annuli in ~70% maximum known size individuals; and IV) uninterrupted laminar tissue in a 100% maximum known size individual, although the possibility that a crack formed along a LAG cannot be eliminated. These results force us to reconsider what an adult-sized specimen might look like for *Lystrosaurus* sp., as the current sample lacks histologic indicators of somatic maturity (e.g., an external fundamental system) [78]. Botha [22] found similar results for South African species of *Lystrosaurus*, suggesting that species typified by small body size (viz. *L. murrayi*) were very likely capable of reaching much larger sizes, but never lived long enough to do so.

Han et al. [40] suggested that sub-adult *Lystrosaurus* from northern Pangea may have reached some level of somatic maturity inferred from parallel-fibered bone and peripheral LAGs in a dorsal rib. However, histological data from the femur, tibia, and fibula of that same individual indicate episodic cessations in growth instead of an overall pattern of slowed growth. The presence of slower growing bone tissue in the ribs likely reflects differential growth rates seen throughout the skeleton, rather than somatic maturity [49, 93].

### Growth marks

Unlike the bone histology reported from South African and Indian *Lystrosaurus*, in which growth marks are rare and often limited to a single LAG or annulus, we find one, and more commonly, two growth marks in immature *Lystrosaurus* from the Jiucaiyuan Formation. The number of growth marks is mostly consistent across associated elements from the same individual as well as across individuals of the same size class (Table 1). Interestingly, the type of growth mark, as either a LAG, a series of closely spaced and occasionally bifurcating LAGs, an annulus consisting of lamellar tissue, or an annulus consisting of parallel-fibered to lamellar tissue, varied throughout the current sample. In most cases, the first (i.e., osteologically deeper) growth mark, or singular growth mark in the case of the humerus, was often recorded as an

annulus accompanied by a reduction of vascular canal sizes. The outer cortical growth marks were more variable, and often varied within the same thin section.

We did not find evidence of growth marks in individuals in Size Class II. These individuals had characteristic juvenile bone histology (i.e., large radial to longitudinal vascular canals that frequently anastomose); therefore our analysis disagrees with the assignment of LAGs in juvenile *Lystrosaurus* by Han et al. [40]: Fig 3. In some cases, the mineralization front can be mis-identified as a growth mark in the context of periosteally deposited bone. Currey [91] and others (e.g., [90]), describe laminar bone formation where accreted circumferential layers of bone (made up of a woven scaffold that is later infilled with parallel-fibered or lamellar bone in the form of primary osteons) is bordered by the periosteum. As the periosteum jumps peripherally when new bone is accreted, a bright line is left between each circumferential layer and represents the place at which the periosteum was located before new bone was added. This mineral-ized line differs from a growth mark as it is not continuous around the cortex and is not hyper-mineralized but appears as a bright line often devoid of osteocyte lacunae [90]. Growth marks, on the other hand, are continuous around the cortex, unless secondary remodeling or cortical remodeling causes discontinuities (i.e., by erosion).

All growth marks observed in our analysis come from Size Class III, which corresponds to individuals that are approximately 70% maximum known size. We can confidently trace up to two growth marks in the current Chinese sample, which is higher than reports from South African and Indian specimens of *Lystrosaurus*, which generally lack growth marks. However, the remainder of the cortical bone tissue in the current Chinese sample reflects the similarly high and consistent rate of bone growth reported from South African and Indian *Lystrosaurus* [22, 54]. A slight increase in the number of growth marks in northern Pangean *Lystrosaurus* could be explained by one or two possible factors that are not mutually exclusive: (i) by an increase in their lifespan or (ii) by environmental stressors.

There is a longstanding hypothesis that the Permo-Triassic extinction caused a Lilliput effect, a persistent reduction in body size, in surviving terrestrial lineages [19, 94–97]. This claim has been investigated histologically in therocephalians, which support an overall reduc-tion in body size following the extinction [19, 97]. A Lilliput effect was also posited for *Lystro-saurus*, based in large part on the overwhelming abundance of small skulls in the Karoo Basin and museum collections in South Africa [20, 21, 98, 99]. However, Botha [22] later rejected a strict Lilliput effect for southern Pangean *Lystrosaurus* (based on fossils from South Africa and India) because of the lack of mature histological tissue in all of the species examined [22]. Instead, she [22] suggested that the large proportion of small skulls in the Triassic was better explained by high mortality in immature individuals, as assessed by their long bone histology.

## Body size

To better understand how *Lystrosaurus* from the Jiucaiyuan Formation compare to congeners from more southerly parts of Pangea, we gathered body size data (based on basal skull length and femur length) from the literature [22, 40, 54] and personal observations of Early Triassic-aged specimens collected in China, South Africa, and India. From this sample, it is clear that Chinese *Lystrosaurus* have a larger average body size (Figs 7 and 8). However, cranial size dis-tributions suggest that both populations from South Africa and China had the potential to grow to similar maximum recorded sizes (Fig 7). The lack of large specimens of *Lystrosaurus* in South Africa aligns with previous interpretations of increased mortality at small sizes by Botha [22]; also see [20]. Our data provide evidence suggesting that northern Pangean *Lystro-saurus* could have reached larger sizes more readily or easily. However, it is unclear the degree to which differences in sampling practices might affect the body size distributions seen here. A

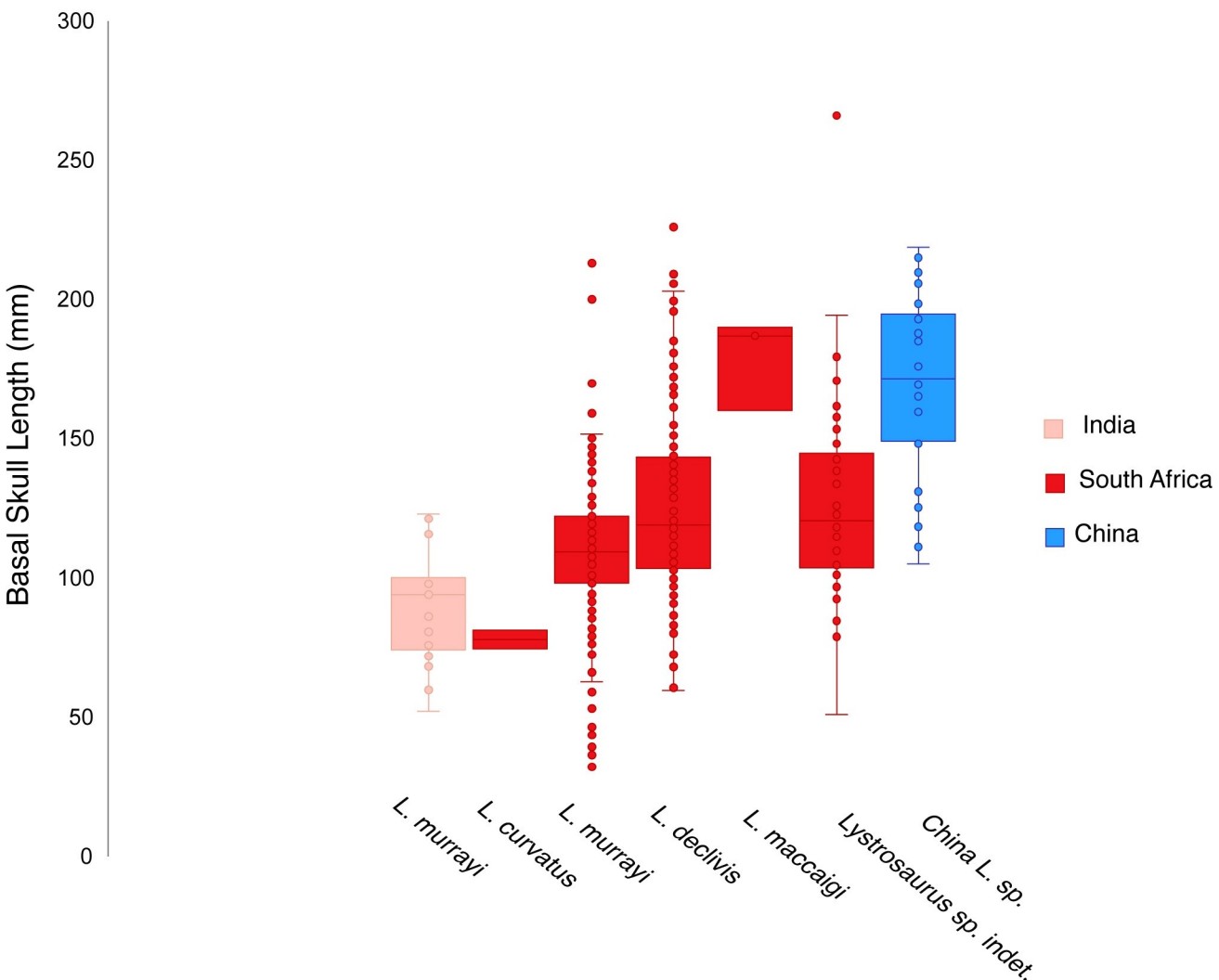

**Fig 7. Interspecific body size distributions from Early Triassic *Lystrosaurus* show an abundance of large individuals from China.** Species from South Africa and China overlap in maximum basal skull length but individuals from China have larger average size. See S1 Dataset for more information.

decade of detailed stratigraphic work targeting the Permo-Triassic boundary in South Africa has led to more comprehensive fossil sampling, which should provide a relatively unbiased view of the vertebrate fauna [100, 101]. However, historical collections from the Karoo likely systematically emphasized rare components of the assemblage (e.g., [102]). Likewise, the Permo-Triassic vertebrate record from Xinjiang has not been subject to the same intensive collecting as that of the Karoo Basin, and therefore it currently may not provide an unbiased view of body size distributions or relative abundances of taxa.

The presence of large individuals with episodic growth marks in all of the subadult-sized elements suggests that *Lystrosaurus* from the Jiucaiyuan Formation were able to reach large sizes through prolonged and rapid growth, unlike Triassic *Lystrosaurus* from South Africa that rarely lived long enough to reach large body sizes despite growing rapidly (Fig 7). There are several possible explanations for this pattern. First, physiological differences among the species of *Lystrosaurus* could be responsible. This proposal is difficult to test, as the alpha-taxonomy of Chinese *Lystrosaurus* is in need of revision and phylogenetic studies of the genus have typically

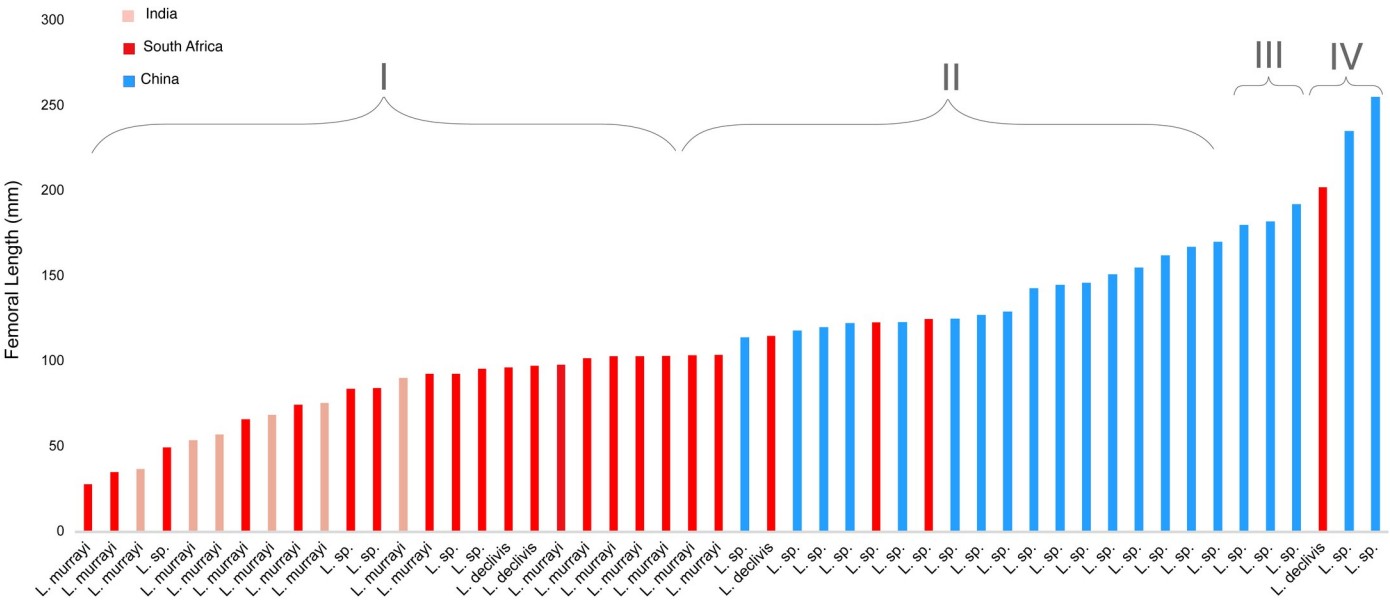

**Fig 8. Interspecific femoral lengths from Early Triassic *Lystrosaurus* show a higher frequency of large sizes from China.** Size Classes denote femora that are less than 40% maximum known size (MKS) in I, between 40–65% MKS in II, approximately 70–75% MKS in III, and greater than 79% in IV. See S1 Dataset for more information.

only included *L. hedini* as a sole northern hemisphere representative (e.g., [103]). However, to the degree that patterns preserved in hard tissue histology can shed light on physiology, the current data suggests that *L. curvatus*, *L. declivis*, *L. maccaigi*, and *L. murrayi* are effectively indistinguishable [22, 31], which indicates a degree of consistency within the genus.

Climatic differences in the high-southern (60˚S) versus mid-northern (45˚N) paleolatitudes [104, 105] is another possible explanation for the observed difference in body size and inferred lifespan in *Lystrosaurus*. Paleoclimate reconstructions of the latest Permian to earliest Triassic show extreme increases in temperature and aridity [106–109], mega monsoons [110, 111], chemical weathering, soil erosion, and geochemical and isotopic signatures of mass wasting and greenhouse conditions [8, 105, 112, 113]. Facies that span the terrestrial Permo-Triassic boundary of the Karoo Basin are interpreted as a change from meandering rivers to low-sinuosity braid-plains, resulting from increased aridity and a die-off of rooted plants that supported river morphologies [114, 115]. However, more recent evidence from isotopic and geochemical data produced differing estimates of the degree to which aridity increased in the Karoo Basin, complicating our understanding of seasonal variation in precipitation and resource availability [111, 116, 117]. Nevertheless, conditions were disruptive enough to have destabilized terrestrial communities and caused a prolonged interval of faunal turnover in southern Pangea during the extinction interval [15, 18, 118].

The extreme environmental and climatic conditions of southern Pangea are not reflected in the paleoclimate reconstructions of the Juicaiyuan Formation [74]. Here, relatively stable sub-humid to semiarid conditions are interpreted from cyclostratigraphic and paleosol analyses in the South Taodonggou study area [74, 75]. Furthermore, fluvial and lacustrine strata host a diversity of plant, root, and wood fossils that are occasionally preserved in situ and show signatures of wildfires during the Early Triassic [74, 119–121]. In sum, evidence from the paleobotanical record of the Jiucaiyuan Formation suggests a lush and vegetated environment that hosted a diverse terrestrial fauna in earliest Triassic times [40, 64]. Taken together, there is mounting evidence from sedimentologic, isotopic, and cyclostratigraphic data that points to

seasonal, semi-arid conditions in the Karoo Basin and stable, sub-humid conditions in the Turpan-Hami Basin.

The extreme environmental and climatic conditions inferred from the Karoo Basin likely truncated *Lystrosaurus* lifespans in the Early Triassic. Similar environmental constraints on maximum size and longevity are also known from Early Triassic therocephalians and terrestrial temnospondyls, which also lack growth marks [19, 52]. By contrast, we propose that the more favorable environmental conditions and vegetated landscape interpreted from the Jiucaiyuan Formation could have hosted longer-lived *Lystrosaurus* during the Early Triassic. Relatively stable environmental conditions could have allowed populations of *Lystrosaurus* to reach larger average body sizes than their southern Pangean relatives. The prevalence of multiple growth marks in some, but not all, immature individuals, further supports developmentally plastic growth in *Lystrosaurus* first proposed for southern Pangean populations [22]. It also suggests that some individuals experienced periodic arrests of growth, perhaps associated with environmental disruptions, but that these disturbances may not have caused widespread mortality on the scale suggested by the South African record. Similarly, some individuals apparently experienced sufficiently stable conditions for them to grow to comparatively large size without interruption. Alternatively, the uninterrupted growth record of the largest specimen sampled here could also be due to the higher stratigraphic position of this individual; it is possible that upper Juicaiyuan rocks represent more favorable environmental conditions. Finally, the absence of peripheral parallel-fibered bone or consistent, circumferential lamellar tissue in the current sample indicates that the maximum size of *Lystrosaurus* from the Jiucaiyuan Formation remains unknown.

## Conclusions

We present bone histology, cranial geometric morphometric analyses, and body size data for *Lystrosaurus* from the Jiucaiyuan Formation of northwestern China and report statistically different cranial morphologies, an extended lifespan, and larger average body size compared to *Lystrosaurus* living in southern Pangea during Early Triassic times. In addition, we report up to two growth marks in immature individuals, which is inconsistent with previous reports of *Lystrosaurus* life histories from the Turpan-Hami Basin [40]. We suggest that it is unlikely that the growth marks reported here represent regular, cyclical patterns and more likely correspond to instances where bone growth stopped due to unfavorable environmental conditions. Indeed, the absence of growth marks in the largest sampled individual indicates an intrinsically high rate of growth that persisted throughout ontogeny. In addition, the presence of two cessations in growth in individuals from Size Class III indicates that smaller, likely more immature, individuals may have been more susceptible to fluctuations in environmental conditions. However, appositional bone growth rebounded to a rapid rate following these cessations, indicating a flexible physiology, as previously proposed for higher-latitude populations of *Lystrosaurus* [22, 31]. A better sample of large limb bones (>250 mm) as well as an analysis of tusk dentine deposition, are needed to clarify what impact the Early Triassic environment may have had on *Lystrosaurus* growth and development in northern Pangea.

## Supporting information

**S1 File. Figures and corresponding tables for cranial geometric morphometric analysis and bone wall thickness measurements.**
(PDF)

**S1 Dataset. Basal skull length measurements (BSL) and femoral length measurements for Early Triassic *Lystrosaurus*.**
(XLSX)

## Acknowledgments

At the IVPP, we thank YI Jian and SHUKANG Zhang for assistance with thin section preparation as well as LIU Jun for access to field data and specimen collections. We also thank Alida Bailleul for helpful discussion and microscope access. Collecting and compiling the entirety of the comparative datasets would not have been possible without the following museum curators, collection managers, colleagues, and friends who we deeply thank: Zaituna Skosan, Claire Browning, Christian Kammerer, Roger Smith, Jennifer Botha, Elize Butler, Bernhard Zipfel, Bruce Rubidge, Sifelani Jirah, Viktor Radermacher, Carl Mehling, and Mark Norell. For their helpful discussion on earlier versions of this manuscript, we thank Bryan Gee, Savannah Olroyd, and Elliott Armour Smith. Helpful comments that greatly improved the quality of this work were given by Jennifer Botha and one anonymous reviewer. Finally, we are indebted to Wan Yang for collecting some of the fossils analyzed here, organizing fieldwork in 2019 and leading the multidisciplinary project that instigated this research.

## Author Contributions

**Conceptualization:** Zoe T. Kulik, Kenneth D. Angielczyk, Christian A. Sidor.

**Data curation:** Zoe T. Kulik.

**Formal analysis:** Zoe T. Kulik, Jacqueline K. Lungmus.

**Funding acquisition:** Kenneth D. Angielczyk, Christian A. Sidor.

**Investigation:** Zoe T. Kulik.

**Methodology:** Zoe T. Kulik, Jacqueline K. Lungmus, Kenneth D. Angielczyk, Christian A. Sidor.

**Project administration:** Zoe T. Kulik.

**Resources:** Kenneth D. Angielczyk, Christian A. Sidor.

**Supervision:** Christian A. Sidor.

**Validation:** Zoe T. Kulik, Kenneth D. Angielczyk, Christian A. Sidor.

**Visualization:** Zoe T. Kulik, Jacqueline K. Lungmus.

**Writing – original draft:** Zoe T. Kulik.

**Writing – review & editing:** Zoe T. Kulik, Jacqueline K. Lungmus, Kenneth D. Angielczyk, Christian A. Sidor.

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
