## [Decision Letter · Decision Letter 0]

12 Jul 2021

PONE-D-21-20672

Living Fast in the Triassic: New data on life history in *Lystrosaurus* (Therapsida: Dicynodontia) from northeastern Pangea

PLOS ONE

Dear Dr. Kulik,

Thank you for submitting your manuscript to PLOS ONE. After careful consideration, we feel that it has merit but does not fully meet PLOS ONE’s publication criteria as it currently stands. Therefore, we invite you to submit a revised version of the manuscript that addresses the points raised during the review process.

We look forward to receiving your revised manuscript.

Kind regards,

Jörg Fröbisch, Ph.D.

Academic Editor

PLOS ONE

Additional Editor Comments:

Dear authors, please pay close attention to the rviewers' comments and suggestions and address them all in a potential revised version. best regards, Jörg Fröbisch

Journal Requirements:

Reviewers' comments:

Reviewer's Responses to Questions

**Comments to the Author**

1. Is the manuscript technically sound, and do the data support the conclusions?

Reviewer #1: No

Reviewer #2: Partly

2. Has the statistical analysis been performed appropriately and rigorously? 

Reviewer #1: N/A

Reviewer #2: Yes

3. Have the authors made all data underlying the findings in their manuscript fully available?

Reviewer #1: Yes

Reviewer #2: Yes

4. Is the manuscript presented in an intelligible fashion and written in standard English?

Reviewer #1: Yes

Reviewer #2: Yes

5. Review Comments to the Author

Reviewer #1: This paper focuses on the osteohistology of Lystrosaurus from the Jiucaiyuan Formation of northwestern China and thus, gives a (relatively) new perspective on Lystrosaurus life history from a northern hemisphere perspective. Lystrosaurus is an ideal study taxon for examining life history changes during mass extinction events as it survived the end-Permian mass extinction to become the most abundant vertebrate in the post-extinction ecosystem. Thus, any new data on Lystrosaurus is globally relevant. This paper is very similar to a recently published paper on Chinese Lystrosaurus osteohistology by Han et al. 2020 even finding very similar conclusions, so the value in this research is not necessarily novel data, but increasing the sample size of Chinese Lystrosaurus to support Han et al.’s 2020 findings. The authors do, however, dispute some of the details of Han et al.’s 2020 paper (which I partially agree with - they may have misidentified a few growth marks, especially in the youngest individuals, but they are correct for the most part) and the authors in this paper provide a very good dataset on the basal skull length of Lystrosaurus skulls from China.

The paper is very well written and detailed and although very similar to the Han et al. 2020 paper, it does increase the sample size of Chinese Lystrosaurus and so should be published once the osteohistological details have been sorted out. There are some issues with identifications and descriptions that need to be fixed or clarified before the paper can be ready for publication.

I have provided numerous comments on the pdf itself, but there are some issues that need to be explained here and corrected in the manuscript.

1. Despite Prondvai et al. 2014 updating bone tissue terminology very few paleohistologists accepted the change in terminology (including myself) until the recent publication of a new osteohistology textbook by Buffrenil et al. 2021. This book finally accepts that the osteohistology terminology should be updated and more correct definitions be attributed to bone tissue types from a developmental perspective. This gives more meaning to the descriptions. The book is very new and only became available during the submission of this paper so it is unlikely the authors have seen it or even have access to it. They are welcome to contact me as I have a pdf copy that I can send them, but this and the Prondvai reference really should be used to describe the bone tissues. I’m not judging, I’ve been remiss myself, but we need to start using the correct terminology now and this will correct or update both the Botha 2020 paper and the Han et al. 2020 paper on what Lystrosaurus bone histology really looks like. I’m sending you a table of definitions (see attachment) that will help you to identify the bone tissue types in the various ontogenetic stages. I can see certain aspects of the extracellular matrix in the high magnification images but neither the figures nor the images on Morphobank are high enough resolution for me to confirm static or dynamic osteogenesis – and you need this to decide what bone tissue type you are looking at. The term fibrolamellar bone is no longer used. Instead bone tissue types have been replaced with woven bone, parallel-fibered bone and the woven-parallel complex. Typically only embryonic or neonates have pure woven bone, which is SO (static osteogenesis) derived. Thus, what you are looking at in the Lystrosaurus bone tissues is the woven-parallel complex, which consists of both SO and DO (dynamic osteogenesis) derived bone tissues. This is where the PFB (parallel-fibered bone) as the lamellae around the primary osteons and within the extracellular matrix is deposited on a scaffold of woven bone. The term fibrolamellar complex (FLC) has been retained to describe tissues that are predominantly woven, but have the PFB component in the form of primary osteons. This is highly vascularized fast growing tissue that you will only see in young juveniles, but falls under the woven-parallel complex. Thus you can still use FLC if you explain what you mean, but from the images I am unsure if you do have FLC in any of your age classes. You may have had it in Age Class I (but you don’t have this), but I am unable to see enough of the osteocytes to see if you have predominantly SO or an equal amount of SO-DO in your Age Class II. In order to prove FLC you will need a very high magnification image of woven bone and it’s possible it’s present in the inner cortex of the bones in Figure 3, but unfortunately Fig 3B and C do not provide enough of the cortex to know if it’s FLC or WPC. These two images show WPC, for example in Fig 3C just under the letter C there is a nodule of woven bone, but unless I saw more of the cortex at this magnification, I cannot tell from your images if you have FLC or WPC. To prove FLC you need to replace Fig 3C with something showing more woven bone. I realise you had a point to 3C, so it might be more useful to replace Fig 3B with tissues showing a higher degree of SO. This follows through all of your age classes – you are not likely to find FLC in your older individuals, even though they are still far from being fully grown, it is likely that you have WPC as the dominant bone tissue type. This is the case for the South African and Indian Lystrosaurus. If it is difficult to redo some of the figures you could add several images to your Morphobank data – high magnification images. The reason why it’s important to get this description right is because, for example, you mention parallel-fibered bone in one of your elements – for an overall change from WPC to PFB, which has not been found in any Lystrosaurus specimens studied to date, would imply that you’ve found an individual that is showing the possible onset of reproductive maturity, or at least that it’s a late subadult, which is older than any Lystrosaurus specimen found to date. I do not see your parallel-fibered bone in the ulna, it still looks like WPC but with a much lower degree of the woven component, i.e. there is still static osteogenesis. The vascular canals in Figure 5M still looks like WPC and not PFB. Thus, if you’re right it has far reaching implications, but you have not shown enough evidence for a change from WPC to PFB. Another reason why it’s important to be absolutely sure you’re dealing with FLC (a subcategory of WPC) is that it indicates very high growth rates compared to WPC (although WPC still indicates high growth rates, depending on the proportion of SO). So if you still have FLC in your older age classes it’s suggesting that these animals were growing faster than Lystrosaurus in other parts of the world or were younger – so again it’s important to differentiate.

Prondvai, E., K. H. Stein, A. de Ricqles and J. Cubo. 2014. Development-based revision of bone tissue classification: the importance of semantics for science. Biological Journal of the Linnean Society 112, 799–816.

Buffrénil, de V., A. de Ricqlès, L. Zylberberg and K. Padian (eds.), Comparative skeletal histology and paleohistology, CRC Press, Taylor and Francis, Boca Raton, 824 pp. (the chapter by Buffrenil and Quilhac on bone tissue types).

I think the paper would benefit from some polarized light images showing the woven look.

2. A very big issue are the vascular orientations. To clear something up that I see in a couple of places in the text, the phrase – laminar to fibrolamellar bone – is incorrect. Laminar bone is a kind of vascular orientation found within the fibrolamellar complex (previously known as FLB), so it IS FLC. I have seen descriptions written like this in papers before, but it is incorrect. The old Francillon et al 1990 book of definitions or the latest osteohistology book by Buffrenil et al 2021 gives you all the definitions of various vascular orientations within the woven-parallel complex. According to the images shown in this paper the younger individuals contain predominantly longitudinally-oriented primary osteons with perhaps some short anastomoses (not enough to be reticular), and as the individuals become older a predominance of plexiform or laminar bone appears. This is important because – although there is an overlap – the rule of thumb is that longitudinal canals indicate faster growth rates than laminar bone – and this would make sense as the older individuals become more laminar. There is also the hypothesis – based on extant studies – that laminar bone is stronger, which you would expect in the older individuals. Most of the vascular orientations for each bone are incorrect and this needs to be corrected. I won’t give you definitions to all the vascular orientations here.

3. The first definition of a growth mark is described as a LAG and annuli are not mentioned until they are described in the description part of the paper, with no definition. Growth marks include both annuli and LAGs (and even changes in vascular orientation), both annuli and LAGs need to be defined when they are first introduced as the bones contain both annuli and LAGs.

4. Lastly I would not suggest you identify the peripheral regions as an EFS in the places mentioned - you can mention it as a possibility but, particularly if you are basing this on ribs – or at least say it is just as likely (more than likely really) that you’re seeing a temporary decrease or cessation in growth. As one is generally dealing with a rib fragment, one doesn’t even know from where along the rib the section is taken, so the histology can change along the bone. You might tend to find more growth marks in ribs, but you really do not know if you’re dealing with an EFS or not – and you make this case in your discussion – that you haven’t found any senescent individuals yet. So I would suggest that you represent these regions as temporary changes in growth, but say that it's not impossible they represent an EFS (so change it around, because it is highly likely you are not looking at an EFS).

Reviewer #2: IVPP V 27127=19SS26 This is a specimen from Shanshan, not Turfan!

It is the only one from the upper portion of the Jiucaiyuan Formation. It should be caution to discuss it. Because it is so different from others!

Also, could IVPP V 26548 represent an adult?

Do you have the original photo of 19SS26, I check the preserved part is less than 20 cm, not 240mm (L241) ?

Abbrev, for IVPP V appears first in Fig 1, anterior to L172

L182 Chang the format as IVPP V 27126.1 not – (a blank after V)

Table 1 For same specimen, you need not add letter behind it.

Fig.1 move below the paragraph begin from L 170

L 30 ‘Formation of northwestern China’ to ‘Formation of Xinjiang, China’

L62 Lystrosaurus has a low preferred temperature than other tetrapods (Liu et al., submitted)

Liu, J., Angielczyk, K.D. and Abdala, F., 2021. Permo-Triassic tetrapods and their climate implications. Global and Planetary Change.

It is almost accepted now.

L109 northwestern to ‘Xinjiang,’

shows uninterrupted cortical growth, suggesting

336 that Lystrosaurus from the Jiucaiyuan Formation had a high intrinsic rate of growth that could

337 periodically arrest.

L702 For Chinese names, you can list like YI Jian, LIU Jun, but all in same sequence

L711 Lystrosaurus change to italicize

6. PLOS authors have the option to publish the peer review history of their article (what does this mean?). If published, this will include your full peer review and any attached files.

Reviewer #1: **Yes: **Jennifer BOTHA

Reviewer #2: No

---

## [Author Response · Author response to Decision Letter 0]

23 Sep 2021

Dear Editors and Reviewers, 

Thank you for your reviews and feedback. We have revised a large portion of the manuscript in response to Reviewer 1's comments and concerns and have included addition stratigraphic information and relevant discussion in response to Reviewer 2. Please find a detailed list of our changes to the text, figures, and supplemental information below.

Kind regards,

Zoe Kulik 

Comments from the Journal

Done

Done

Comments from Reviewer 1:

Reviewer #1: This paper focuses on the osteohistology of Lystrosaurus from the Jiucaiyuan Formation of northwestern China and thus, gives a (relatively) new perspective on Lystrosaurus life history from a northern hemisphere perspective. Lystrosaurus is an ideal study taxon for examining life history changes during mass extinction events as it survived the end-Permian mass extinction to become the most abundant vertebrate in the post-extinction ecosystem. Thus, any new data on Lystrosaurus is globally relevant. This paper is very similar to a recently published paper on Chinese Lystrosaurus osteohistology by Han et al. 2020 even finding very similar conclusions, so the value in this research is not necessarily novel data, but increasing the sample size of Chinese Lystrosaurus to support Han et al.’s 2020 findings. The authors do, however, dispute some of the details of Han et al.’s 2020 paper (which I partially agree with - they may have misidentified a few growth marks, especially in the youngest individuals, but they are correct for the most part) and the authors in this paper provide a very good dataset on the basal skull length of Lystrosaurus skulls from China.

The paper is very well written and detailed and although very similar to the Han et al. 2020 paper, it does increase the sample size of Chinese Lystrosaurus and so should be published once the osteohistological details have been sorted out. There are some issues with identifications and descriptions that need to be fixed or clarified before the paper can be ready for publication.

I have provided numerous comments on the pdf itself, but there are some issues that need to be explained here and corrected in the manuscript.

1. Despite Prondvai et al. 2014 updating bone tissue terminology very few paleohistologists accepted the change in terminology (including myself) until the recent publication of a new osteohistology textbook by Buffrenil et al. 2021. This book finally accepts that the osteohistology terminology should be updated and more correct definitions be attributed to bone tissue types from a developmental perspective. This gives more meaning to the descriptions. The book is very new and only became available during the submission of this paper so it is unlikely the authors have seen it or even have access to it. They are welcome to contact me as I have a pdf copy that I can send them, but this and the Prondvai reference really should be used to describe the bone tissues. I’m not judging, I’ve been remiss myself, but we need to start using the correct terminology now and this will correct or update both the Botha 2020 paper and the Han et al. 2020 paper on what Lystrosaurus bone histology really looks like. I’m sending you a table of definitions (see attachment) that will help you to identify the bone tissue types in the various ontogenetic stages. I can see certain aspects of the extracellular matrix in the high magnification images but neither the figures nor the images on Morphobank are high enough resolution for me to confirm static or dynamic osteogenesis – and you need this to decide what bone tissue type you are looking at. The term fibrolamellar bone is no longer used. Instead bone tissue types have been replaced with woven bone, parallel-fibered bone and the woven-parallel complex. Typically only embryonic or neonates have pure woven bone, which is SO (static osteogenesis) derived. Thus, what you are looking at in the Lystrosaurus bone tissues is the woven-parallel complex, which consists of both SO and DO (dynamic osteogenesis) derived bone tissues. This is where the PFB (parallel-fibered bone) as the lamellae around the primary osteons and within the extracellular matrix is deposited on a scaffold of woven bone. The term fibrolamellar complex (FLC) has been retained to describe tissues that are predominantly woven, but have the PFB component in the form of primary osteons. This is highly vascularized fast growing tissue that you will only see in young juveniles, but falls under the woven-parallel complex. Thus you can still use FLC if you explain what you mean, but from the images I am unsure if you do have FLC in any of your age classes. You may have had it in Age Class I (but you don’t have this), but I am unable to see enough of the osteocytes to see if you have predominantly SO or an equal amount of SO-DO in your Age Class II. In order to prove FLC you will need a very high magnification image of woven bone and it’s possible it’s present in the inner cortex of the bones in Figure 3, but unfortunately Fig 3B and C do not provide enough of the cortex to know if it’s FLC or WPC. These two images show WPC, for example in Fig 3C just under the letter C there is a nodule of woven bone, but unless I saw more of the cortex at this magnification, I cannot tell from your images if you have FLC or WPC. To prove FLC you need to replace Fig 3C with something showing more woven bone. I realise you had a point to 3C, so it might be more useful to replace Fig 3B with tissues showing a higher degree of SO. This follows through all of your age classes – you are not likely to find FLC in your older individuals, even though they are still far from being fully grown, it is likely that you have WPC as the dominant bone tissue type. This is the case for the South African and Indian Lystrosaurus. If it is difficult to redo some of the figures you could add several images to your Morphobank data – high magnification images. The reason why it’s important to get this description right is because, for example, you mention parallel-fibered bone in one of your elements – for an overall change from WPC to PFB, which has not been found in any Lystrosaurus specimens studied to date, would imply that you’ve found an individual that is showing the possible onset of reproductive maturity, or at least that it’s a late subadult, which is older than any Lystrosaurus specimen found to date. I do not see your parallel-fibered bone in the ulna, it still looks like WPC but with a much lower degree of the woven component, i.e. there is still static osteogenesis. The vascular canals in Figure 5M still looks like WPC and not PFB. Thus, if you’re right it has far reaching implications, but you have not shown enough evidence for a change from WPC to PFB. Another reason why it’s important to be absolutely sure you’re dealing with FLC (a subcategory of WPC) is that it indicates very high growth rates compared to WPC (although WPC still indicates high growth rates, depending on the proportion of SO). So if you still have FLC in your older age classes it’s suggesting that these animals were growing faster than Lystrosaurus in other parts of the world or were younger – so again it’s important to differentiate.

Prondvai, E., K. H. Stein, A. de Ricqles and J. Cubo. 2014. Development-based revision of bone tissue classification: the importance of semantics for science. Biological Journal of the Linnean Society 112, 799–816.

Buffrénil, de V., A. de Ricqlès, L. Zylberberg and K. Padian (eds.), Comparative skeletal histology and paleohistology, CRC Press, Taylor and Francis, Boca Raton, 824 pp. (the chapter by Buffrenil and Quilhac on bone tissue types).

I think the paper would benefit from some polarized light images showing the woven look.

2. A very big issue are the vascular orientations. To clear something up that I see in a couple of places in the text, the phrase – laminar to fibrolamellar bone – is incorrect. Laminar bone is a kind of vascular orientation found within the fibrolamellar complex (previously known as FLB), so it IS FLC. I have seen descriptions written like this in papers before, but it is incorrect. The old Francillon et al 1990 book of definitions or the latest osteohistology book by Buffrenil et al 2021 gives you all the definitions of various vascular orientations within the woven-parallel complex. According to the images shown in this paper the younger individuals contain predominantly longitudinally-oriented primary osteons with perhaps some short anastomoses (not enough to be reticular), and as the individuals become older a predominance of plexiform or laminar bone appears. This is important because – although there is an overlap – the rule of thumb is that longitudinal canals indicate faster growth rates than laminar bone – and this would make sense as the older individuals become more laminar. There is also the hypothesis – based on extant studies – that laminar bone is stronger, which you would expect in the older individuals. Most of the vascular orientations for each bone are incorrect and this needs to be corrected. I won’t give you definitions to all the vascular orientations here.

3. The first definition of a growth mark is described as a LAG and annuli are not mentioned until they are described in the description part of the paper, with no definition. Growth marks include both annuli and LAGs (and even changes in vascular orientation), both annuli and LAGs need to be defined when they are first introduced as the bones contain both annuli and LAGs.

4. Lastly I would not suggest you identify the peripheral regions as an EFS in the places mentioned - you can mention it as a possibility but, particularly if you are basing this on ribs – or at least say it is just as likely (more than likely really) that you’re seeing a temporary decrease or cessation in growth. As one is generally dealing with a rib fragment, one doesn’t even know from where along the rib the section is taken, so the histology can change along the bone. You might tend to find more growth marks in ribs, but you really do not know if you’re dealing with an EFS or not – and you make this case in your discussion – that you haven’t found any senescent individuals yet. So I would suggest that you represent these regions as temporary changes in growth, but say that it's not impossible they represent an EFS (so change it around, because it is highly likely you are not looking at an EFS).

We have read the suggested publications and chapters from the recent textbook and have added a Bone Histology Terminology section to the Materials and Methods section to add additional information relevant to the recent changes in terminology. We have also updated the histological descriptions throughout and added high magnification images to Figs 3, 5, and 6 to address the majority of Reviewer 1’s questions and comments. Details of our changes are explained below. 

Line 79: annuli also represent annual changes in growth and you need to include them here too, even if your samples only show LAGs, the SA samples also have annuli, so you need to explain to the reader that you are talking about growth marks (which includes LAGs and annuli)

The Osteohistological Perspectives on Life History section has been expanded to include descriptions and definitions for both annuli and LAGs. 

Line 92: you need to refer to growth marks (general term) here - not just LAGs, as it's the scarcity of annuli AND LAGs. Here you are only referring to one type of growth mark, annuli are still deemed annual

Corrected for clarity

Table 1 Column 5: this is the first time you mention growth marks - you haven't defined this in the text. This is why you need to define growth marks - not just LAGs when you first mention them

Updated definitions in Osteohistological Perspectives on Life History section

Line 323: how did you measure this? This is important for comparisons as you will notice that Fig 3F, all in Fig 4, Fig 5 and fig 6 have a gradual transition between medullary cavity and compact cortex. One needs to be consistent in defining where the edge of the medullary cavity is, which is not possible in Lystrosaurus (juding it by eye). You need to explain exactly how you measured these sections - from open medullary cavity, from where there was 50% more bone than cavity space, from where you deemed the cortex to be compact (which can be subjective). You need to show a diagram, at least, if you going to do it by eye. Otherwise, in order to do it properly you need to use ImageJ or Bone Profiler to get K and thus the cortical thickness

Based on your recommendation, we have replaced the bone wall thickness measurements with cortical thickness (K). This resulted in some updated measurements and corrections of clerical errors to Table 1. We attempted to use BoneJ to determine the relative porosity in the gradual transition of trabecular medullary infilling to cortical bone but dark minerals that infilled the medullary and vascular spaces resulted in unreliable values. We proceeded to use a visual estimation of the transition to compact cortical bone and include two supplemental figures that illustrate where our measurements were taken (S1 Supplementary Information SF5 and SF6). 

Line 326: I will give you a detailed explanation in my overall comments as to why I've changed the details here to reflect more up to date terminology

Thank you for these detailed comments. 

Line 326: you're mixing up vascular canal orientation - laminar is a type of vascular orientation, you can have laminar or plexiform bone but you cannot have laminar bone with a plexiform orientation

Edited for clarity 

Line 330: this is why you need to define growth marks earlier - this is the first time you mention annuli and you don't define what they are

Updated to include both, see above 

Line 357: you need to prove that the extracellcular matrix has a woven component. Looks for areas between the osteons - outside of the PFB lamellae that have bunched, large rounded osteocyte lacunae. They will look like nodules or bunches. You need to show a high magnification image of these osteocytes to prove this is not PFB. you could use 3C and point to the area just below the C - otherwise in B at the lower left corner there is some woven bone, otherwise you can take a new image

Added text labels to Fig 3B and replaced one of the images with a high-magnification image illustrating woven-fibered bone in an associated fibula (Fig 3E). 

Line 361: rephrase - an annulus is not a growth arrest, but a growth decrease, you can say growth cycles

Rephrased 

Line 363: none of these elements are showing laminar or plexiform tissue (you see it in the older individuals but not here), here the dominant vascularization is longitudinally-oriented primary osteons in circumferential rows, with some short anastomoses - but they're not laminar or plexiform - redo the vascularization here

Done

Line 365: you need to say this is the yellow square in A. 

Done

Line 365: be careful here - your arrows aren't actually pointing to woven bone. I understand what you're meaning here and you are correct, but the arrows are just pointing to areas where the osteocyte lacunae are not preserved - and may be woven or parallel-fibred. In 3C there is a woven patch just under the letter C in the image - but your arrows are pointing to something we do not know. A woven-parallel complex has both woven and parallel components

Edited figure caption and accompanying text. We refer to these lines as bright lines in the text and have updated the figure caption for clarity. 

Line 369: no same as D - this is not plexiform

This image has been replaced to show a high magnification image of woven-fibered bone matrix.

Line 381: no - see my comments in your Figure 3 caption. Yes you are seeing fast growth but this is a woven-fibered complex, and there is no plexiform bone here

Edited

Line 385: This band seems to run around most of the cortex and is likely a temporary decrease in growth rate (and you need to introduce the possibility that it could be annual). Growth marks needn't be narrow annuli of PFB of LB, but they are can also be represented by changes in vascular canal orientation or a temporary decrease in vascular size - this has to mean a temporary decrease in growth rate

With respect to the smaller vascular canals in the proximal tibia, we have edited the text to refer to this temporary decrease in canal sizes as a temporary shift in growth rate. We do not interpret the differences in vascular canal size or orientations as temporary decreases in growth rate in various quadrants of the fibula and maintain that the different tissue textures seen in Fig 3A are not indicative of shifting bone growth rates in annual or cyclical patterns. 

Line 388: and what is the woven versus parallel-fibered component - is there more PFB in the organized region and more woven bone in the less organized region, or does the proportion remain the same and it's only the vascular canal orientation that changes? I can't see from the Figure

Edited to add a description of the proportion of woven-fibered bone in this element. 

Line 393: you need to add here why they shouldn't be mistaken for annuli either - i.e. if you don't see a cement line, why shouldn't you call them annuli? so you need to point out that these bright lines do not contain bands of slower forming tissues like parallel-fibered or lamellar bone. 

Good point. We have added more to the description. 

Line 393: not reticular - you need to check your definitions, I'll explain in the overall comments - Your Stage II mostly comprises longitudinally-oriented primary osteons in circumferential rows - Fig 3A, 3F. Definitely not enough anastomoses to call this reticular and the canals are not circumferentially arranged (for laminar or plexiform). Longitudinal canals arranged circumferentially is different to canals arranged circumferentially around the bone

‘Reticular’ and ‘plexiform’ have been replaced with ‘longitudinally-oriented primary osteons’ throughout. 

Line 407: no - laminar is a kind of vascularization that forms part of the traditional fibrolamellar complex. You cannot have laminar to fibrolamellar bone - this is comparing apples to oranges. The vascularization is not truely laminar yet like you get in Figure 5J. The canals are a mixture of sub-laminar and longitudinally-oriented primary osteons in circumferential rows

Edited for clarity. It is our preference to refer to this tissue as a mixture of longitudinal and circular canals that approaches a laminar organization. 

Line 408: what bone tissue makes up the annulus? PFB or LB?

Edited to add: parallel-fibered. (None of the osteocyte lacunae are lenticular-shaped so it seems these lacunae were cut orthogonally) 

Line 414: hind limb

Done

Line 414: check these orientations - you haven't been accurate about the previous descriptions and the resolution of the renders are not high enough for me to see clearly what's happening in E and H - but F and G look sub-reticular not plexiform . There are longitudinal canals at the subperiosteal surface of F and I and L

We have updated the vascular orientation to reflect differences in each of the 3 elements, rather than try to combine the differences in one sentence. 

Line 420: then it's probably the PFB part of the WPC where woven bone is becoming discontinuous

Edited to replace lamellar with parallel-fibered bone. 

Line 425: This definitely looks like a LAG in the inner yellow bracket - how far does it go? I cannot see if the lines are cement lines or lamellae - but either way if they go most of the way around the cortex they are likely growth marks and should be pointed out in this image

Updated figure caption and accompanying text. The thickness of the avascular PFB region is pretty variable and the subperiosteal edge is not complete around the entire section, but the parallel-fibered tissue goes most of the way around the cortex and is certainly a growth mark. 

Line 427: it's a double LAG - there's no bone tissue between them and they likely represent one season

Agreed. 

Line 429: The image isn't quite high resolution enough - but it looks like what you're seeing is the PFB component of the WPC increasing towards the outer cortex. As these are older individuals you should see a progressive decrease in the amount of woven bone and an increase in PFB towards the outer cortex. If that outer region in K goes around the whole cortex then it's likely a growth mark (annulus)

Yes, there is a shift towards more organized parallel-fibered bone in some regions of the outermost cortex, but it is likely not an annulus. In the vascularized regions (4L), the mineralized lines wrap around primary osteons rather than cross-cutting, so we are interpreting these lines as the mineralizing front of periosteally deposited bone. 

Line 450: you need a higher magnification of Fig 5I showing the "multiple stacked LAGs". The current image shows a cycle from too low magnification so one cannot see that there are LAGs - from this it looks like an annulus. If you zoom in on your yellow bracket more you can show them properly

Edited to refer to growth mark as an annulus. 

Line 454: no take this out - you've just said in the next sentence what it most likely represents. These animals are no way near fully grown

Done

Line 456: exactly

Thank you

Line 472: now this is proper laminar vascularization - do you see how the canals are more connected to one another circumferentially compared to the younger elements where the vascular canals are more longitudinal with less connections?

Agreed

Line 480: are you absolutely sure there isn't one closer to the medullary cavity? You don't show a close up so I can't tell from the overall render, it's either just a change in colour or it could be an annulus - just double check

It is just a color change. 

Line 482: laminar bone IS fibrolamellar bone. Yes it's laminar but that refers to the vascularization. You would call a woven-parallel complex with laminar vascularization. I'll explain in my overall review

Edited

Line 483: no - they are laminar, just like you said

Edited

Line 492: Are you absolutely sure you are only seeing dynamic osteogenesis and there is no static osteogenesis? You need to be really sure about this before you call it PFB - this has huge implications, because none of your other elements show this and neither do the South African samples, so if you're right you seeing the beginning of what is probably reproductive maturity - but you have to be absolutely sure - this is very important

We have updated the figures to show a region in the outer cortex that reflects the overall tissue type that is largely parallel-fibered but has small areas of woven-fibered bone in the ulna. The osteocyte lacunae are less densely packed than in the hind limb elements but do show areas of static osteogenesis. 

Line 496: no but if you're seeing an overall change from the woven-parallel complex to parallel-fibered bone then you could be seeing reproductive maturity. But the amount of woven bone decrease dramatically with age, but if there are still patches of it, then it's not PFB like you've said above. I'm afraid I cannot see the osteocytes well enough in M to help you here, and you'd need to check around the whole of the outer cortex

The change is not dramatic enough to suggest maturity, rather, there is just less woven-fibered bone 

Line 501: This bone shows continued rapid growth but is bigger and presumably older than the ulna in Size Class III - which is why I'm hesitant to call that PFB in the ulna. You need to check this as it has important repercussions. If you're right then it suggests a decoupling of size and reproductive maturity. It is more likely that the ulna is woven-parallel all the way to the edge

Edited above (see line 492)

Line 503: OK - so this makes no sense. The inner cortex represents younger ontogeny and shouldn't have been growing slower than the outer cortex which represents the older ontogeny. This all looks like woven-parallel complex, where you'll have varying degrees of woven versus parallel-fibred bone.

The birefringence under cross-polarized light was really difficult to diagnose. After looking more closely at the osteocyte lacunae, we see woven- and parallel-fibered bone throughout the middle and outer cortex. There seems to be more parallel-fibered bone in the outer cortex compared to the deep and middle cortex, but not an overall shift from one tissue type to another. 

Line 511: no it doesn't - and if it's laminar vascularization this is not going to be PFB but rather WPC

Corrected, see above 

Line 515: you don't know this - as you've seen, a thick slower growing region at the sub-periosteal surface is likely a seasonal annulus. Yes it's not impossible it's an EFS, but if you nor Han nor myself have ever found fully grown individuals of Lystrosaurus - what is more likely? That this individual represents a fully grown animal, or that it represents a temporary decrease in growth rate? You need to put both possibilities out there and the latter is far more likely

We have removed references to an EFS and have added to the description to explain the uncertainty raised by your comments.

Line 517: same comment - you don't know this is an EFS

We have removed references to an EFS. 

Line 526: I don't see a stack of LAGs - I see a region of PFB that may or may not represent an EFS - you really need to rather think of the more likely possibility (as explained above) that these bones do not represent fully grown individuals. The only way we will find definitive evidence of fully grown Lystrosaurus is if we find it in the limb bones. I really would not base this on the ribs, as I've seen many ribs like this where the associated limb bones show that the animal is still growing

Perhaps the emphasis was lost here. We agree that ribs alone, and especially fragmentary ones, do not indicate whether an animal is fully grown since ribs grow at different rates than limbs. We have edited the description to explain the uncertainty raised by your comment. 

Line 534: I disagree with your vascular orientations - see earlier comments

Edited throughout

Line 537: I would call this more laminar

Edited

Line 558: no - an annulus cannot consist of stacked LAGs. I was wondering what you were meaning by stacked LAGs in the description. There's no high enough magnification for me to confirm - but what I think youré actually seeing is an annulus consisting of lamellar tissue - and the "stacked LAGs"are the individual lamellae. You can have an annulus and a LAG together - where the annulus occurs before or after the LAG - but you cannot have an annulus consisting of multiple LAGs. If you are indeed seeing LAGs and not lamellae, you need to show a high magnification image of this, so far I haven't seen any evidence of this

Edited throughout

Line 558: avascular what? tissue? lamellar tissue? parallel-fibered tissue? It can be avascular but what tissue type is it?

Edited to add: parallel-fibered to lamellar tissue

Line 568: you need to cite Buffrenil et al 2021 for this - they give a detailed account (first given by de Ricqles) regarding the "unit". Woven bone is deposited first as a scaffold on which parallel-fibered bone (in the form of primary osteons) is deposited. So it's not really that bone is deposited first and then woven bone is deposited - it's more the other way around, this is why you need this book. You can also look at Prondvai et al 2014

The type of bone deposition we are referring to here is the accretion of periosteal bone, rather than the formation of primary osteons. Currey (1960) and others (e.g., Mori et al., Cells Tissues Organs. 2003) describe laminar bone formation where each subsequent circumferential layer of bone (made up of a woven scaffold that is later infilled with PFB or LB) is bordered by the periosteum. As the periosteum jumps peripherally when new bone is accreted, a bright line is left between each circumferential layer and represents the place at which the periosteum was located before new bone was deposited. 

601: I'm not sure I understand your meaning of maximum sizes here. We don't know what the maximum size of any of these species was, so how can one say they could grow to similar MAXIMUM sizes? Please clarify

Edited to add: maximum recorded size 

Line 656: no it doesn't - unless you can provide evidence that you get plants/trees that take multiple years to reach maturity and do not produce growth rings? Otherwise isn't this more likely that the wood fossils represent immature plants that are less than a year old. If you are correct please provide references that show that trees can grow multiple years without laying down growth rings.

We have removed this statement based on updated reports of fossil wood with growth marks and charcoal from the study area. 

Wan M-L, Yang W, Wan S, Wang J. Wildfires in the Early Triassic of northeastern Pangaea: Evidence from fossil charcoal in the Bogda Mountains, northwestern China. Palaeoworld. 2021 Jul

Line 668: agreed

No change 

Line 711: italix

Done

Comments from Reviewer 2:

IVPP V 27127=19SS26 This is a specimen from Shanshan, not Turfan!

It is the only one from the upper portion of the Jiucaiyuan Formation. It should be caution to discuss it. Because it is so different from others!

Also, could IVPP V 26548 represent an adult?

Do you have the original photo of 19SS26, I check the preserved part is less than 20 cm, not 240mm (L241)?

We have updated stratigraphic information for V 27127. It does come from higher in the section which we have added to the discussion section as a possible explanation for why the bone tissue is different from some of the smaller specimens. We have an original photo with a scale bar from the field for 19SS26 and, while incomplete, it is over 25cm long. For specimen IVPP V 26548, we do not see evidence of somatic maturity (i.e., EFS) which makes it extremely unlikely that this individual was an adult. 

Abbrev, for IVPP V appears first in Fig 1, anterior to L172

We don’t believe we need an explanation of the abbreviation in this figure based on the explanation currently in the text.

L182 Chang the format as IVPP V 27126.1 not – (a blank after V)

Table 1 For same specimen, you need not add letter behind it.

We changed the formatting of the specimen numbers throughout. Our preference is to keep the letter following each element from associations of the same specimen for clarity. 

Fig.1 move below the paragraph begin from L 170

Change Size Distribution Figure to FIG 2. Put it after Geometric Morphometric Fig

Our preference is to keep the limb size distribution as Fig 1 since the emphasis of this paper is on the bone histology of that sample that is first introduced in the Introduction section. 

L 30 ‘Formation of northwestern China’ to ‘Formation of Xinjiang, China’

Done

L62 Lystrosaurus has a low preferred temperature than other tetrapods (Liu et al., submitted)

Liu, J., Angielczyk, K.D. and Abdala, F., 2021. Permo-Triassic tetrapods and their climate implications. Global and Planetary Change.

It is almost accepted now.

This manuscript was still in review at the time we submitted the manuscript. However, a revised version has been resubmitted, and we now cite the paper and note that the unusual inferred thermal tolerances of Lystrosaurus could have contributed to its survival. 

L109 northwestern to ‘Xinjiang,’

Done

shows uninterrupted cortical growth, suggesting

336 that Lystrosaurus from the Jiucaiyuan Formation had a high intrinsic rate of growth that could

337 periodically arrest.

No change

L702 For Chinese names, you can list like YI Jian, LIU Jun, but all in same sequence

Done, thank you

L711 Lystrosaurus change to italicize

Done

---

## [Decision Letter · Decision Letter 1]

14 Oct 2021

PONE-D-21-20672R1Living Fast in the Triassic: New data on life history in *Lystrosaurus* (Therapsida: Dicynodontia) from northeastern PangeaPLOS ONE

Dear Dr. Kulik,

Thank you for submitting your manuscript to PLOS ONE. After careful consideration, we feel that it has merit but does not fully meet PLOS ONE’s publication criteria as it currently stands. Therefore, we invite you to submit a revised version of the manuscript that addresses the points raised during the review process.

We look forward to receiving your revised manuscript.

Kind regards,

Jörg Fröbisch, Ph.D.

Academic Editor

PLOS ONE

Journal Requirements:

Additional Editor Comments:

Dear Authors,

I consider the manuscript essentially publishable as is, but I would like you to have a look at the latest comments by the one reviewer. Please see, if you would like to address the raised points or not and briefly reply to them (order of the figures, nomenclature, crack=LAG? etc.). Once you resubmit your comments and a potential revised version, I'll be happy to accept the manuscript for publication and pass that recommendation on to the editor in chief(editorial office.

Best, Jörg

Reviewers' comments:

Reviewer's Responses to Questions

**Comments to the Author**

1. If the authors have adequately addressed your comments raised in a previous round of review and you feel that this manuscript is now acceptable for publication, you may indicate that here to bypass the “Comments to the Author” section, enter your conflict of interest statement in the “Confidential to Editor” section, and submit your "Accept" recommendation.

Reviewer #1: All comments have been addressed

2. Is the manuscript technically sound, and do the data support the conclusions?

Reviewer #1: Yes

3. Has the statistical analysis been performed appropriately and rigorously? 

Reviewer #1: N/A

4. Have the authors made all data underlying the findings in their manuscript fully available?

Reviewer #1: Yes

5. Is the manuscript presented in an intelligible fashion and written in standard English?

Reviewer #1: Yes

6. Review Comments to the Author

Reviewer #1: I'm very happy with this second version. The authors have either made the corrections I suggested, or clarified my questions/confusion with clearer images. The authors are still a little wary of referring to dominant bone tissue type as a woven-parallel complex, and instead refer to it as both woven and parallel-fibered - I'm ok with this, it just might be confusing for less experienced readers who might think the authors are referring to overall parallel-fibered bone and not WPC. But I'll let the authors decide what they want to do. I have very minor comments on the pdf attached. Just something to check, I noticed figure 4 was placed after figure 6 in the document. One more thing, the largest femur might in fact have a LAG - where the crack is. I've noticed that cracks often run along LAGs - I can't prove to you that it's a potential growth mark, so I'll leave it to the authors to decide.

Otherwise I'm happy for the MS to be published as is - very nice piece of work, well done!

7. PLOS authors have the option to publish the peer review history of their article (what does this mean?). If published, this will include your full peer review and any attached files.

Reviewer #1: **Yes: **Jennifer Botha

---

## [Author Response · Author response to Decision Letter 1]

15 Oct 2021

Dear Editors, 

We have addressed the comments and suggestions raised by Reviewer 1 and have edited the manuscript to reflect those changes. In particular, we add the possibility that the crack in the largest femur may have formed along a LAG. Based on the section that was added on bone tissue terminology that addresses our choice of terminology, we do not wish to make any additional changes. Finally, figures should appear in numerical order from 1 through 8. Please find a list of the changes that were made below. 

Kind regards,

Zoe Kulik

Reviewer #1: I'm very happy with this second version. The authors have either made the corrections I suggested, or clarified my questions/confusion with clearer images. The authors are still a little wary of referring to dominant bone tissue type as a woven-parallel complex, and instead refer to it as both woven and parallel-fibered - I'm ok with this, it just might be confusing for less experienced readers who might think the authors are referring to overall parallel-fibered bone and not WPC. But I'll let the authors decide what they want to do. I have very minor comments on the pdf attached. Just something to check, I noticed figure 4 was placed after figure 6 in the document. One more thing, the largest femur might in fact have a LAG - where the crack is. I've noticed that cracks often run along LAGs - I can't prove to you that it's a potential growth mark, so I'll leave it to the authors to decide.

Otherwise I'm happy for the MS to be published as is - very nice piece of work, well done!

Line 77: Replaced ‘slowdowns’ with ‘decreases’ 

Done

Line 403: Replace pose with propose?

Done

Line 453: Add space between full stop and next word

Done

Line 501: change to LAGs

Edited for clarity 

Line 578: that crack runs suspiciously consistently around the same area of cortex around the whole bone. I often see cracks running along LAGs, especially in dinosaurs, where the LAG cracks along it's whole circumference - it's not impossible that the crack represents a LAG

Edited to add more description of the cracks and the possibility that the circumferential crack formed along a LAG 

Line 595: Replace lamellar with parallel-fibered

Edited throughout rib description and figure caption

Line 609: It would be interesting to note here that the thickness of the cortex in each rib differs, casting doubt of Ray et al.'s 2005 hypothesis that Lystrosaurus had exceptionally thick ribs, which they suggested was indicative of an aquatic lifestyle. You can clearly see from your sample that it really does depend on where the section is taken

Good point, we added this comment here and, in the discussion, as suggested 

Line 619: although it is possible that the femur has a growth mark as mentioned above

Added an additional statement about this uncertainty

---

## [Editor Report · Decision Letter 2]

19 Oct 2021

Living Fast in the Triassic: New data on life history in *Lystrosaurus* (Therapsida: Dicynodontia) from northeastern Pangea

PONE-D-21-20672R2

Dear Dr. Kulik,

We’re pleased to inform you that your manuscript has been judged scientifically suitable for publication and will be formally accepted for publication once it meets all outstanding technical requirements.

Kind regards,

Jörg Fröbisch, Ph.D.

Academic Editor

PLOS ONE
---

## [Editor Report · Acceptance letter]

28 Oct 2021

PONE-D-21-20672R2 

Living Fast in the Triassic: New data on life history in *Lystrosaurus* (Therapsida: Dicynodontia) from northeastern Pangea 

Dear Dr. Kulik:

I'm pleased to inform you that your manuscript has been deemed suitable for publication in PLOS ONE. Congratulations! Your manuscript is now with our production department. 

Kind regards, 

on behalf of

Prof. Jörg Fröbisch 

Academic Editor

PLOS ONE